EMBO
Molecular Medicine

# Deficiency in intestinal epithelial O-GlcNAcylation predisposes to gut inflammation

Ming Zhao[1,2,†], Xiwen Xiong[1,†], Kaiqun Ren[2,3], Bing Xu[4], Meng Cheng[2], Chinmayi Sahu[2], Kaichun Wu[4], Yongzhan Nie[4,‡], Zan Huang[2,5,6], Richard S Blumberg[7], Xiaonan Han[8,9] & Hai-Bin Ruan[1,2,*] [iD]

## Abstract

Post-translational modifications in intestinal epithelial cells (IECs) allow for precise control in intestinal homeostasis, the breakdown of which may precipitate the pathological damage and inflammation in inflammatory bowel disease. The O-linked β-N-acetylglucosamine (O-GlcNAc) modification on intracellular proteins controls diverse biological processes; however, its roles in intestinal homeostasis are still largely unexplored. Here, we found that levels of protein O-GlcNAcylation and the expression of O-GlcNAc transferase (OGT), the enzyme adding the O-GlcNAc moiety, were reduced in IECs in human IBD patients. Deletion of OGT specifically in IECs resulted in disrupted epithelial barrier, microbial dysbiosis, Paneth cell dysfunction, and intestinal inflammation in mice. Using fecal microbiota transplantation in mice, we demonstrated that microbial dysbiosis although was insufficient to induce spontaneous inflammation but exacerbated chemical-induced colitis. Paneth cell-specific deletion of OGT led to Paneth cell dysfunction, which might predispose mice to chemical-induced colitis. On the other hand, the augmentation of O-GlcNAc signaling by inhibiting O-GlcNAcase, the enzyme removing O-GlcNAcylation, alleviated chemical-induced colitis. Our data reveal that protein O-GlcNAcylation in IECs controls key regulatory mechanisms to maintain mucosal homeostasis.

**Keywords** epithelial barrier function; gut microbiota; inflammatory bowel disease; Paneth cells; STAT signaling
**Subject Categories** Digestive System; Immunology; Post-translational Modifications, Proteolysis & Proteomics

## Introduction

Inflammatory bowel disease (IBD), including ulcerative colitis (UC) and Crohn's disease (CD), is a group of conditions characterized by chronic or recurring inflammation of the gastrointestinal tract. The induction and perpetuation of intestinal inflammation require the convergence of several abnormalities that affect overlapping layers of homeostatic modules including genetic predisposition, barrier dysfunction, microbial dysbiosis, and immune over-activation (Khor et al, 2011; Maloy & Powrie, 2011; Kayama & Takeda, 2012; Kamada et al, 2013; Knights et al, 2013; Sonnenberg & Artis, 2015). Intestinal epithelial cells (IECs) establish a barrier between luminal environment and the internal milieu, placing IECs at the center of interactions between the mucosal immune system and luminal antigens and metabolites. A healthy and robust layer of IECs maintains multiple layers of intestinal homeostasis. Dysfunction in IEC biology, such as epithelial barrier malfunction, uncontrolled cell death, and defective autophagy in Paneth cells, drives intestinal inflammation (Khor et al, 2011; Gilbert et al, 2012; Peterson & Artis, 2014).

The epithelial barrier is primarily mediated by the formation of junction complexes between IECs, which connect adjacent IECs to form a continuous physical barrier that restricts luminal pathogens from invading the intestine. The turnover of IECs, such as apoptosis, provides an additional challenge to the maintenance of epithelial continuity. In IBD, dysregulation of junction complexes and cell death both contribute to the "leaky gut" and intestinal inflammation (Turner, 2009).

The Paneth cell is a type of secretory IECs found at the base of the small intestine crypt. It contains large granules high in antimicrobial peptides (AMPs), which can alter the composition of gut microbiota and counteract enteric pathogens (Bevins & Salzman, 2011). A recent study has demonstrated Paneth cells as a site of

1 School of Forensic Medicine, Xinxiang Medical University, Xinxiang, Henan, China
2 Department of Integrative Biology and Physiology, University of Minnesota Medical School, Minneapolis, MN, USA
3 College of Medicine, Hunan Normal University, Changsha, Hunan, China
4 State Key Laboratory of Cancer Biology & Institute of Digestive Diseases, Xijing Hospital, The Fourth Military Medical University, Xi'an, Shaanxi, China
5 Laboratory of Gastrointestinal Microbiology, Jiangsu Key Laboratory of Gastrointestinal Nutrition and Animal Health, College of Animal Science and Technology, Nanjing Agriculture University, Nanjing, Jiangsu, China
6 National Center for International Research on Animal Gut Nutrition, Nanjing Agriculture University, Nanjing, Jiangsu, China
7 Division of Gastroenterology, Department of Medicine, Brigham and Women's Hospital, Harvard Medical School, Boston, MA, USA
8 Division of Gastroenterology, Hepatology, and Nutrition, Cincinnati Children's Hospital Medical Center, Cincinnati, OH, USA
9 MOH Key Laboratory of Human Disease Comparative Medicine, Institute of Laboratory Animal Science, Chinese Academy of Medical Science (CAMS) and Peking Union Medical College (PUMC), Beijing, China
  *Corresponding author. Tel: +1 612 301 7686; Fax: +1 612 301 1229; E-mail: hruan@umn.edu
  †These authors contributed equally to this work
  ‡Correction added on 7 August 2018 after first online publication: Yongzan Nie was corrected to Yongzhan Nie.

origin for intestinal inflammation (Adolph *et al*, 2013). Several genetic susceptibility alleles for human IBD, such as *ATG16L1*, *NOD2*, and *XBP1*, all lead to Paneth cell dysfunction (Bevins & Salzman, 2011; Khor *et al*, 2011). Taken together, IECs control multi-layers of intestinal homeostasis, the disruption of which make a major contribution to the IBD pathogenesis.

O-GlcNAcylation is the post-translational modification of serine and threonine residues with β-N-acetylglucosamine (O-GlcNAc) on intracellular proteins (Torres & Hart, 1984; Hart *et al*, 2007). This dynamic modification is attached by O-GlcNAc transferase (OGT) and removed by O-GlcNAcase (OGA). Protein O-GlcNAcylation acts as a hormone and nutrient sensor to control many biological processes including cell signaling, metabolism, development, and aging (Hanover *et al*, 2012; Ruan *et al*, 2012, 2013b, 2014; Yang & Qian, 2017). Nevertheless, the role of intestinal epithelial O-GlcNA-cylation in barrier function and inflammation is still largely unexplored. Herein, we found that levels of OGT and protein O-GlcNAcylation were downregulated in IECs of IBD patients. IEC-specific knockout of OGT in mice resulted in permeable epithelial barrier, Paneth cell dysfunction, microbial dysbiosis, and ultimately intestinal inflammation. Elevating intestinal O-GlcNAcylation levels increased barrier function and protected mice from chemical-induced inflammation. Our data demonstrate that protein O-GlcNA-cylation in IECs is important for the intestinal homeostasis.

# Results

### Defective O-GlcNAc modification in IECs in IBD patients

To explore the potential involvement of O-GlcNAc modification in intestinal inflammation, we performed the immunohistochemistry staining of OGT and protein O-GlcNAcylation on colon tissues of IBD patients (Appendix Table S1). In both UC and CD, levels of OGT and O-GlcNAc modification were robustly downregulated in IECs, compared to those in controls (Fig 1A and B). We also observed a similar reduction in levels of OGT and O-GlcNAcyla-tion in a separate cohort of Chinese UC patients (Fig EV1A and B). Although not statistically significant, the expression of *OGT* gene in UC patients tended to decrease when compared to healthy subjects (Fig EV1C). To further evaluate whether there was any correlation between disease severity and levels of OGT and O-GlcNAcylation, we performed the immunohistochemistry staining on another set of intestine biopsies (Appendix Table S2) and confirmed the reduction in OGT and O-GlcNAcylation levels in IECs of both UC and CD (Fig 1C). Interestingly, intensities of

both OGT and O-GlcNAcylation were negatively correlated with histological disease activities in UC and CD that were determined by Geboes and global histological activity (GHA) scores, respectively (Fig 1D and E, and Dataset EV1; van Loosdregt & Coffer, 2014). These data demonstrate that O-GlcNAc dysfunction in IECs is associated with IBD.

### Epithelial deficiency in OGT causes intestinal damages in mice

To directly examine the role of protein O-GlcNAcylation in the intestinal epithelia, we generated IEC-specific *Ogt* gene knockout mice (*Vil-Ogt* KO) by crossing the *Villin-Cre* and *Ogt-floxed* mouse lines. Immunohistochemistry demonstrated that OGT and O-GlcNAcylation were specifically and efficiently depleted in both ileum and colon in *Vil-Ogt* KO mice (Fig EV2A and B). Male *Vil-Ogt* KO mice were viable, but substantially lighter in weight (Fig 2A) and gradually developed rectal prolapse, rectal bleeding, and diarrhea (Fig 2B and C). In females, heterozygous KO mice appeared normal and healthy, while homozygous KO females showed similar phenotypes as KO males including the body weight loss and the progressive rectal prolapse (Fig 2D and E). Later, we used male mice for most experiments. Histological analyses showed that intestinal architecture was disrupted in *Vil-Ogt* KO mice, such as the irregularity of the size and shape of crypts and increased crypt branching (Fig 2F).

We then utilized semi-quantitative pathology to qualify the pathological alterations in ileal and colonic mucosa (Erben *et al*, 2014; Gilbert *et al*, 2015). Significantly increased intestinal epithelial hypertrophy, epithelial hyperplasia, and mucosal thickness were observed in the *Vil-Ogt* KO mice (Fig 3A and B). Immunohistochemistry revealed a modest increase in infiltrating neutrophils, macrophages, and CD4 T cells at the crypt base of the ileum in *Vil-Ogt* KO mice (Fig 3C). Quantitative reverse transcription PCR (RT–qPCR) showed that the expression of inflammatory genes including *Tnfa*, *Il6*, *Il1b*, and *Ifng* was largely upregulated in the jejunum, ileum, and colon of *Vil-Ogt* KO mice when compared to control mice (Fig 3D). Immunofluorescence of Ki-67 showed significantly greater amounts of proliferating epithelial cells in the ileum of *Vil-Ogt* KO (Fig 3E). In addition, TUNEL assay and immunostaining of Cleaved-CASPASE3 showed increased apoptotic cells in the crypt region of ileum in *Vil-Ogt* KO mice (Fig 3F and G). Consistently, cultured ileal organoids from KO mice had a profound decrease in viability, when compared to control organoids (Fig 3H). Collectively, our data illustrate that loss of protein O-GlcNAcylation in IECs results in dysregulated proliferative/apoptotic homeostasis and susceptibility to inflammation in mice.

---

**Figure 1. Defective O-GlcNAc signaling in intestinal epithelial cells in IBD patients.**

A, B  Representative images of OGT (A) and O-GlcNAc (B) immunohistochemistry on paraffin-embedded colon sections from control, UC, and CD patients ($n = 4$). Intensities of staining were scored on the right. Scale bars = 50 μm. (One-way ANOVA, OGT: Healthy versus UC $P = 0.0002$, Healthy versus CD $P = 0.003$; O-GlcNAc: Healthy versus UC $P = 0.0365$, Healthy versus CD $P = 0.0125$).

C      Levels of intestinal epithelial OGT and O-GlcNAcylation in a second cohort of control, UC, and CD patients. (One-way ANOVA, OGT: Control $n = 10$, UC $n = 9$, CD $n = 6$; Control versus UC $P = 0.035$, Control versus CD $P = 0.003$; O-GlcNAc: Control $n = 10$, UC $n = 8$, CD $n = 8$; Control versus UC $P = 0.0027$, Control versus CD $P = 0.008$).

D, E  Regression plots of average immune-staining scores of OGT and O-GlcNAcylation against histological scores in UC (D) and CD (E) (D: OGT $n = 19$, O-GlcNAc $n = 18$; E: OGT $n = 15$, O-GlcNAc $n = 16$).

Data information: Data are represented as mean ± SEM. *$P < 0.05$; **$P < 0.01$; ***$P < 0.001$ by one-way ANOVA with the Dunnett *post hoc* test.

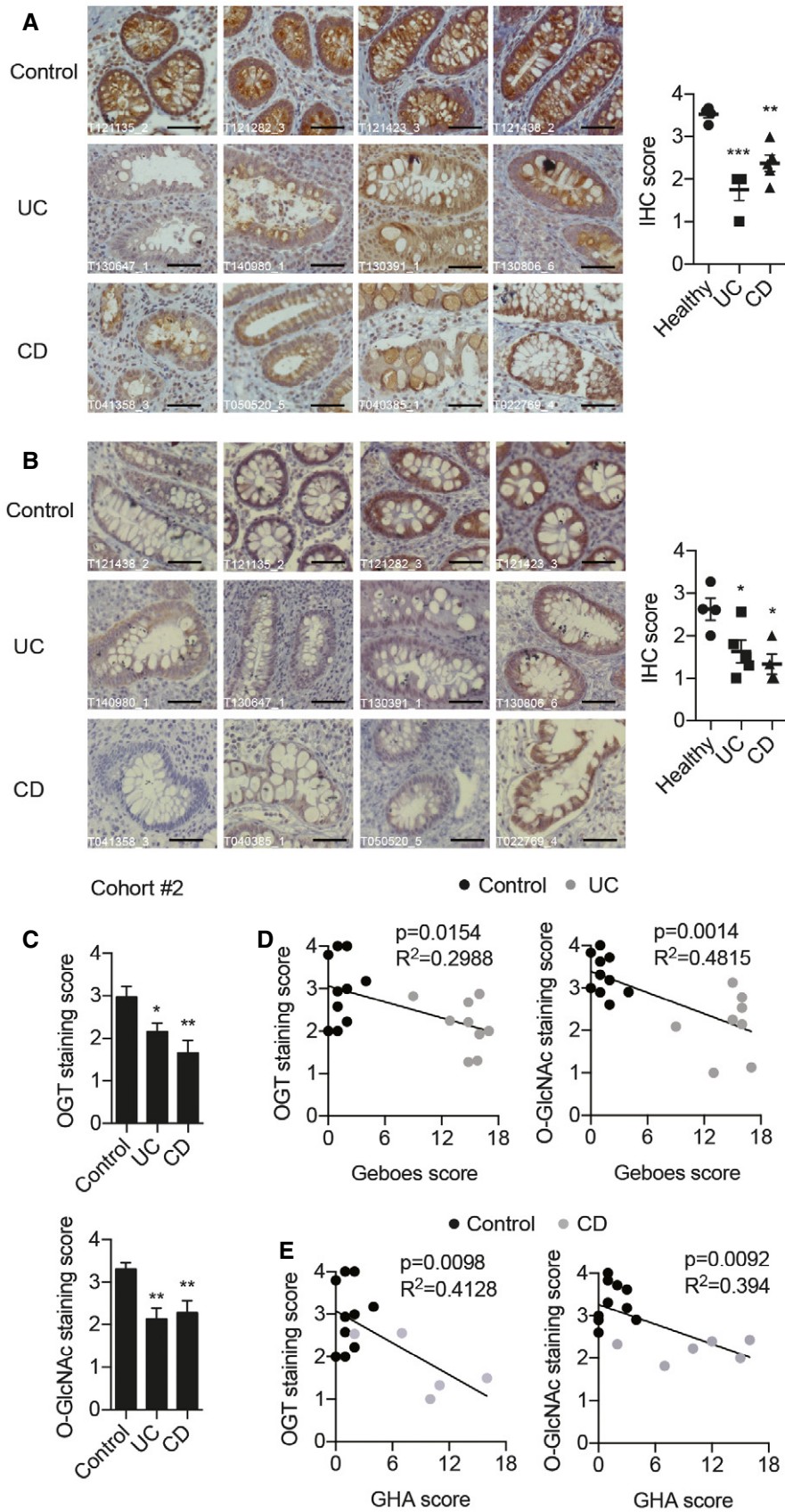

**Figure 1.**

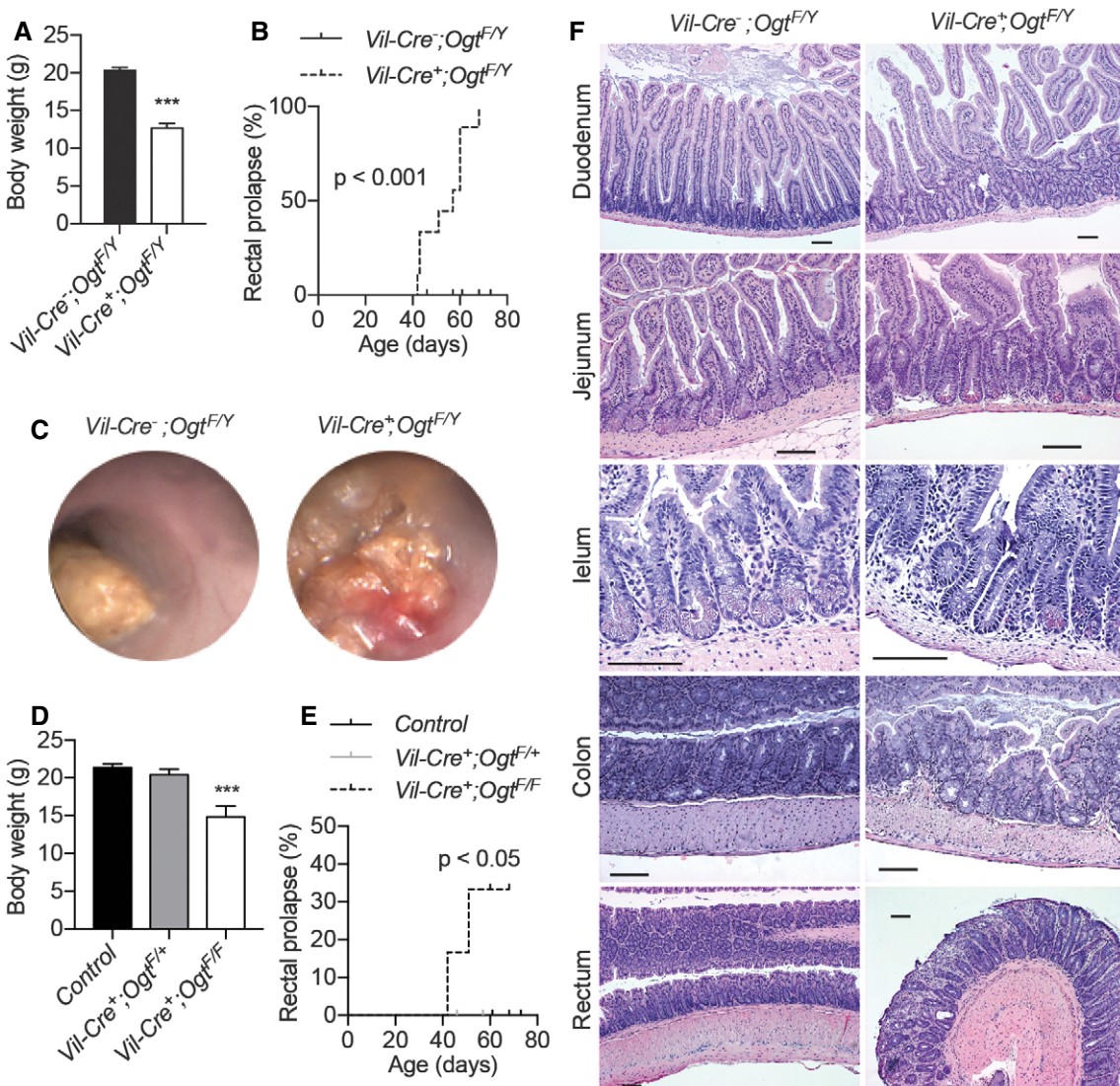

**Figure 2. Loss of OGT in IECs causes intestinal damages in mice.**

A   Body weight of male wild-type and *Vil-Ogt* KO mice at 6 weeks of age (*n* = 6, *P* = 0.000015).
B   Incidence of rectal prolapse in male wild-type and *Vil-Ogt* KO mice (WT *n* = 17, KO *n* = 9, *P* < 0.0001).
C   Representative colonoscopy images of male wild-type and *Vil-Ogt* KO mice.
D   Body weight of female wild-type and *Vil-Ogt* KO mice at 9 weeks of age (WT *n* = 11, heterozygous KO *n* = 4, homozygous KO *n* = 5, *P* < 0.0001).
E   Incidence of rectal prolapse in female wild-type and *Vil-Ogt* KO mice (WT *n* = 9, heterozygous KO *n* = 8, homozygous KO *n* = 6, *P* = 0.0439).
F   H&E staining of duodenum, jejunum, ileum, colon, and rectum of 10-week-old male wild-type and *Vil-Ogt* KO mice. Scale bars = 100 μm.

Data information: Data are represented as mean ± SEM. ***P < 0.001 by two-tailed *t*-test (A), Mantel–Cox test (B and E), or one-way ANOVA with the Dunnett *post hoc* test (D).

## Defective intestinal barrier in *Vil-Ogt* KO mice

Intestinal barrier dysfunction potentiates and sustains intestinal inflammation (Turner, 2009), and we then sought to test whether the deficiency in O-GlcNAcylation in IECs alters epithelial barrier function. *Vil-Ogt* KO mice exhibited increased plasma levels of fluorescein isothiocyanate (FITC) after oral gavage of FITC-dextran (Fig 4A) and increased fecal albumin (Fig 4B), indicating that *Vil-Ogt* KO mice had increased intestinal permeability. A major component of the mucosal barrier is the junction complex between

IECs (Turner, 2009). Electron microscopy of the wild-type intestine clearly showed the tight junctions (arrows), the adherens junctions (arrowheads), and desmosomes (stars) (Fig 4C). In *Vil-Ogt* KO mice, however, the density of perijunctional ring was profoundly decreased (Fig 4C). Desmosomes were reduced in number, which sometime caused open paracellular space (Fig 4C). We also observed splenomegaly (Fig EV3A) and increased levels of inflammatory gene expression in the liver (Fig EV3B), indicating that the leaky gut in *Vil-Ogt* KO mice caused systemic inflammation.

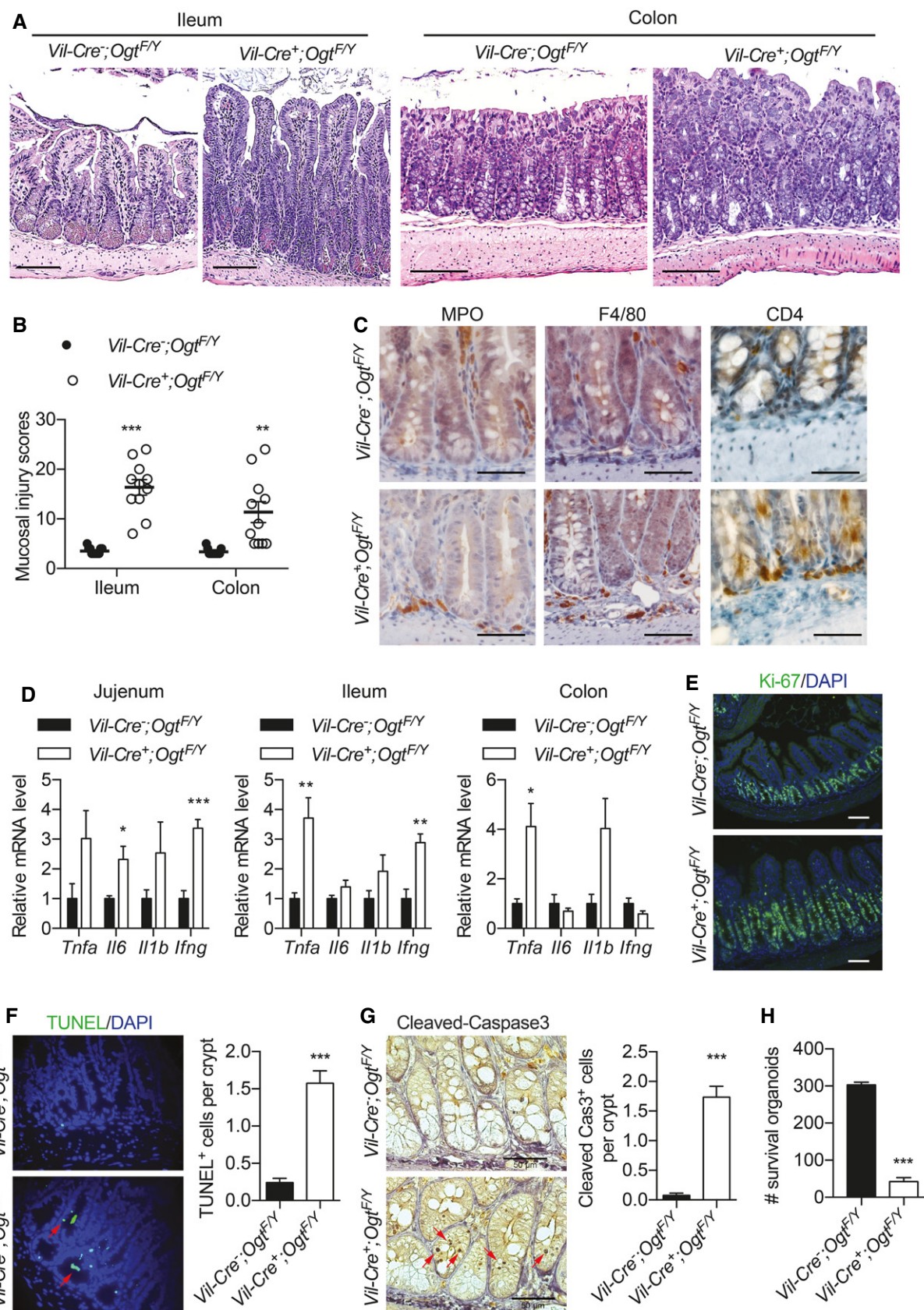

**Figure 3.**

**Figure 3. IEC-specific loss of OGT disrupts intestinal proliferative/apoptotic homeostasis.**

A Representative images of the H&E staining of ileum and colon of 10-week-old male wild-type and *Vil-Ogt* KO mice. Scale bars = 100 μm.

B Combined scores of the mucosal injury in 10-week-old male wild-type and *Vil-Ogt* KO mice (*n* = 11, ileum: *P* < 0.0001, colon: *P* = 0.0012).

C Representative images of MPO, F4/80, and CD4 immunohistochemistry in the ileum tissue of male wild-type and *Vil-Ogt* KO mice. Scale bars = 50 μm.

D RT–qPCR of inflammatory markers in jejunum, ileum, and colon (*n* = 5, jejunum: Il6 *P* = 0.029, *Ifng P* = 0.0009; ileum: *Tnfa P* = 0.0094, *Ifng P* = 0.0066; colon: *Tnfa P* = 0.028).

E Immunofluorescent staining of Ki-67 in ileal tissues (*n* = 5). Scale bars = 50 μm.

F, G TUNEL staining (F) and immunostaining of Cleaved-CASPASE3 (G) in the ileum. Quantification of the numbers of positive cells is shown at the right. (TUNEL WT *n* = 9, KO *n* = 6, *P* = 0.0003; Cleaved Cas3 WT = 8, KO = 7, *P* < 0.0001).

H Numbers of survived organoids per 1000 isolated ileal crypts (WT *n* = 4, KO *n* = 23, *P* < 0.0001).

Data information: Data are represented as mean ± SEM. **P* < 0.05; ***P* < 0.01; ****P* < 0.001 by two-tailed *t*-test.

We then performed Western blotting and immunofluorescence using antibodies against markers of junction complexes and found that PLAKOGLOBIN, a critical component of both desmosomes and adherens junctions that anchors transmembrane cadherins to intermediate filaments, was downregulated in *Vil-Ogt* KO mice (Fig 4D–F). Even though the expression level was not changed, the tight junction protein zonula occludens-1 (ZO-1) remarkably lost its localization to the cell membrane in *Vil-Ogt* KO mice (Fig 4D and F). We did not observe any changes in other junction complex markers including ZO-2, β-CATENIN, α-E-CATENIN, DESMOPLAKIN, CLAUDIN 3 and 5, and OCCLUDIN (Figs 4D and F, and EV3C). Taken together, these findings demonstrate that the loss of OGT in IECs impairs epithelial barrier function, which may potentiate the development of intestinal inflammation.

## Microbial dysbiosis in *Vil-Ogt* KO mice

Microbial dysbiosis plays an important role in inflammatory bowel disease (Bevins & Salzman, 2011), and we then sought to determine whether the deficiency in intestinal epithelial O-GlcNAcylation interferes with gut microbiota. 16S rRNA gene sequencing was used to interrogate differences in fecal microbiota between singly housed wild-type and *Vil-Ogt* KO mice. The phylogenetic diversity was comparable between control and KO mice (Fig EV4A). Principal coordinate analysis (PCoA) of the unweighted UniFrac showed that bacterial communities within the same genotype had similar bacterial composition no matter where they were housed (Fig 5A). Linear discriminant analysis (LDA) effect size (LEfSe) highlighted differentially abundant taxonomic clades (Fig EV4B). The phylum *Firmicutes*, the *Coriobacteriia* class within the *Actinobacteria* phylum, and *Mycoplasmatales* order within the *Tenericutes* phylum were decreased, while the *Bacteriodetes* family *Bacteriodaceae* and the *Gammaproteobacteria* class within the *Proteobacteria* phylum were enriched in *Vil-Ogt* KO mice (Figs 5B and EV4C and D). Interestingly, the most consistent observations of microbial dysbiosis in IBD patients are a reduction in *Firmicutes* and an increase in *Proteobacteria* (Matsuoka & Kanai, 2015).

To determine whether microbial dysbiosis in *Vil-Ogt* KO mice contributes to the pathogenesis of intestinal inflammation, we first performed fecal microbiota transplantation (FMT) from control and *Vil-Ogt* KO mice into antibiotic-treated wild-type mice. Compositional differences between the microbiota from control and *Vil-Ogt* KO mice generally persisted in the corresponding FMT recipients (Fig EV4E). No difference in intestinal morphology (Fig EV4F) or barrier function (Fig EV4G and H) was observed between mice transplanted with microbiota from wild-type or *Vil-Ogt* KO mice. We

then induced acute colitis in these mice with dextran sodium sulfate (DSS) and found that mice receiving microbiota from *Vil-Ogt* KO mice showed a tendency to lose more body weight (Fig 5C), increased colitis score (Fig 5D), and increased intestinal permeability at 5 days after DSS removal (Fig 5E). Histological examination showed severe intestinal injury and inflammation in *Vil-Ogt* KO FMT mice after 5-day recovery (Fig 5F), characterized by severer immune infiltration, more necrotic crypts, and larger size of ulcers when compared to the control group (Fig 5G). The expression of inflammatory markers including *Il1b* and *Il6* in recipients of *Vil-Ogt* KO microbiota showed an increasing tendency as compared to mice undergoing a control FMT (Fig 5H). These data together indicate that, although not sufficient to induce intestinal dysfunction, microbial dysbiosis in *Vil-Ogt* KO mice exacerbates DSS-induced colitis.

## OGT regulates Paneth cell function

The Paneth cell is a type of secretory IECs found at the base of the small intestine crypt. It contains large granules high in anti-microbial peptides (AMPs), which can alter the composition of gut microbiota and counteract enteric pathogens (Bevins & Salzman, 2011). Recent studies have demonstrated Paneth cells as a site of origin for intestinal inflammation (Adolph *et al*, 2013). We found that Paneth cells were missing in some crypts in the ileum of *Vil-Ogt* KO mice (Figs 3A and 6A). H&E staining and electron microscopy further showed that KO Paneth cells had much smaller but more eosinophilic granules (Figs 3A and 6B). TUNEL and Cleaved-CASPASE3 staining was striking in the crypt base where Paneth cells are located (Fig 3F and G). However, the expression of the *Lzy1* gene, a marker of Paneth cells, was not changed (Fig 6C), suggesting that OGT knockout might not affect the differentiation of Paneth cells but rather promote Paneth cell death. The expression of AMP genes, including various *Defensins* and *Ang4*, was significantly downregulated in *Vil-Ogt* KO ileum (Fig 6C). Mounting evidence suggests that IBD-associated mutations in autophagy-related genes impair Paneth cell function (Cadwell *et al*, 2009; Henckaerts *et al*, 2011; Patel & Stappenbeck, 2013; Chu *et al*, 2016). Consistent with our recent finding that protein O-GlcNAcylation promotes hepatic autophagy (Ruan *et al*, 2017), pro-LC3 was accumulated and failed to be cleaved and activated to form LC3-II in the IEC of *Vil-Ogt* KO mice (Fig 6D). These data indicate that loss of OGT causes Paneth cell dysfunction.

To determine whether O-GlcNAcylation autonomously controls Paneth cell function, we generated Paneth cell-specific OGT (*Defa6-Ogt*) KO mice by crossing the *Ogt-floxed* mice to the *Defa6-iCre* line (Adolph *et al*, 2013). Immunofluorescent staining demonstrated that

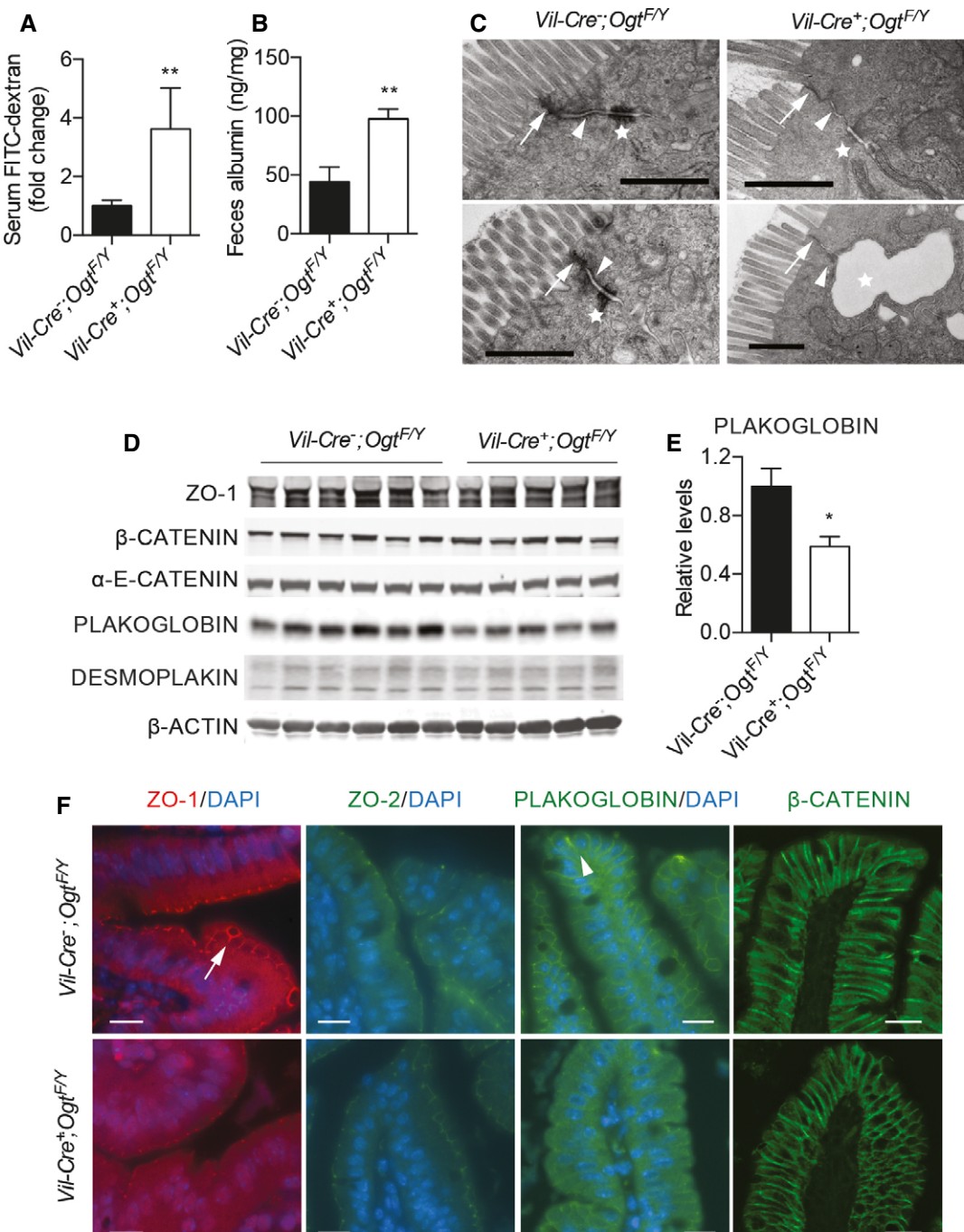

**Figure 4.    Intestinal barrier dysfunction in *Vil-Ogt* KO mice.**

A, B    Intestinal barrier functional assays measuring serum FITC-dextran (A) and albumin from fecal sample (B) of male wild-type and *Vil-Ogt* KO mice. (A: WT *n* = 5, KO *n* = 6, *P* = 0.0015; B: WT *n* = 9, KO *n* = 6, *P* = 0.0053).

C    Representative EM pictures of ileum epithelial cells. Arrows, arrowheads, and stars indicate tight junctions, adherens junctions, and desmosomes, respectively. Scale bars = 1 μm.

D    Immunoblotting of protein markers of tight junction (ZO-1), adherens junction (β-CATENIN, α-E-CATENIN, and PLAKOGLOBIN), and desmosome (PLAKOGLOBIN and DESMOPLAKIN) in colon.

E    Densitometric analysis of PLAKOGLOBIN protein levels in (D). (WT *n* = 6, KO *n* = 5, *P* = 0.021).

F    Representative images of ZO-1, ZO-2, PLAKOGLOBIN, and β-CATENIN immunofluorescent staining in the ileum tissue of male wild-type and *Vil-Ogt* KO mice. Arrow and arrowhead indicate ZO-1 and PLAKOGLOBIN, respectively. Scale bars = 10 μm.

Data information: Data are represented as mean ± SEM. **P* < 0.05; ***P* < 0.01 by two-tailed *t*-test.
Source data are available online for this figure.

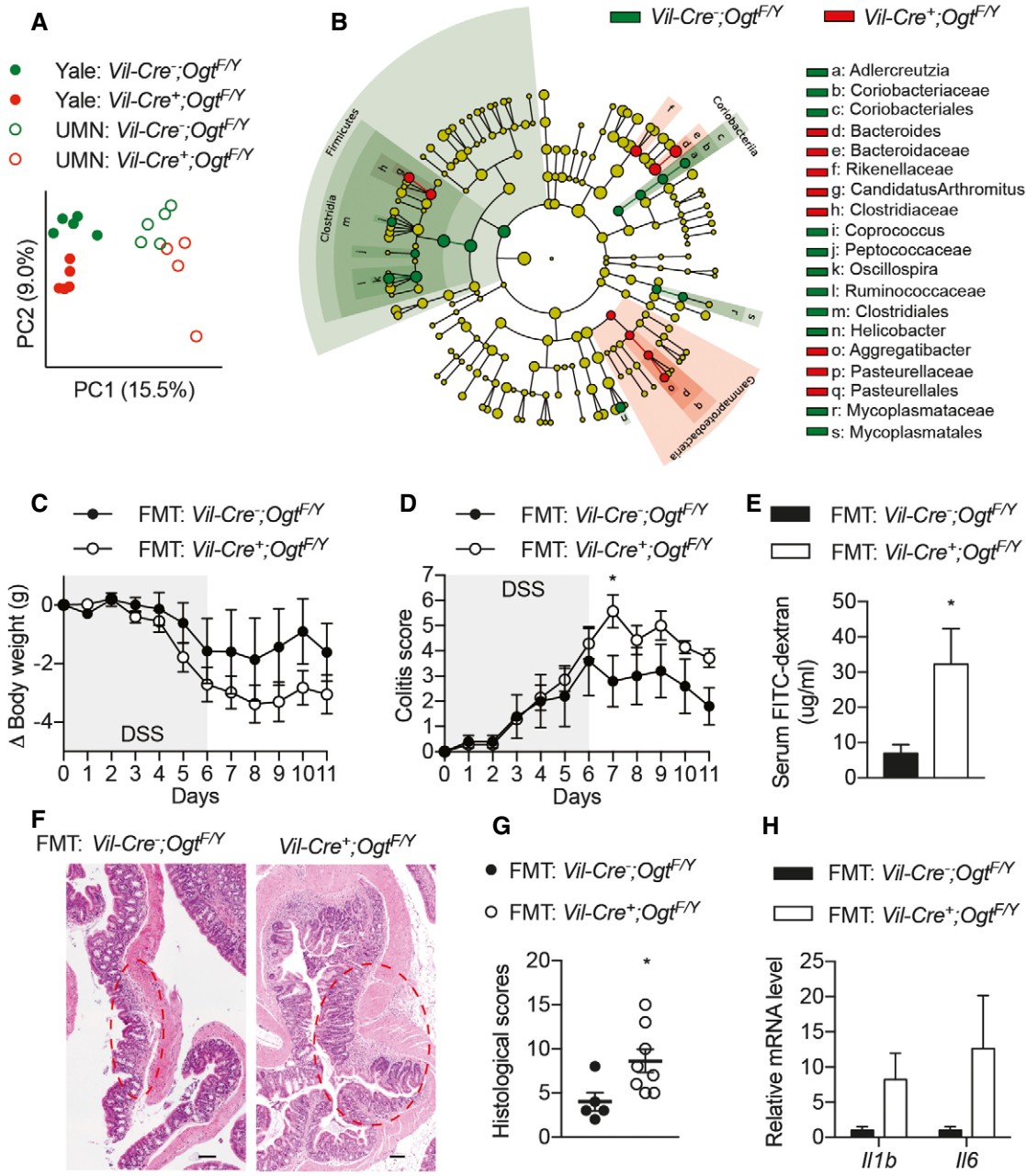

**Figure 5. Microbial dysbiosis in OGT-deficient mice.**

A    PCoA plot of unweighted UniFrac distance of bacterial communities in wild-type and *Vil-Ogt* KO mice that were singly housed at Yale or UMN (UMN KO $n = 4$, rest $n = 5$).

B    Cladogram showing the most discriminative bacterial clades between wild-type and *Vil-Ogt* KO mice that were identified by LEfSe. Regions in red indicate clades that were enriched in *Vil-Ogt* KO mice, while regions in green indicate clades that were enriched in wild-type mice.

C–H    Antibiotic-treated C57BL/6 male mice were transplanted with fecal microbiota from wild-type or *Vil-Ogt* KO mice and then induced with DSS for colitis (WT $n = 5$, KO $n = 7$). (C) Daily changes in body weight. (D) Colitis scores. (E) Intestinal barrier functional assays measuring serum FITC-dextran on Day 10. (F) H&E staining of colon tissues. Ulcer areas are designated by red circles. Scale bars = 200 μm. (G) Pathological scores of the mucosal injury. (H) RT–qPCR of inflammatory markers in the colon. (D: $P = 0.0409$; E: $P = 0.0457$; G: $P = 0.032$).

Data information: Data are represented as mean ± SEM. *$P < 0.05$ by two-way ANOVA followed by Bonferroni corrections (D) and two-tailed *t*-test (E and G).

O-GlcNAcylation was enriched in the nucleus of Paneth cells in WT ileum, and it was specifically and efficiently depleted in Paneth cells of *Defa6-Ogt* KO mice (Fig 6E). H&E staining showed that these mutant mice had fewer Paneth cells and granules were more eosinophilic, similar to the morphological changes observed in *Vil-Ogt* KO mice (Fig 6F). The reduction in Paneth cell number was confirmed

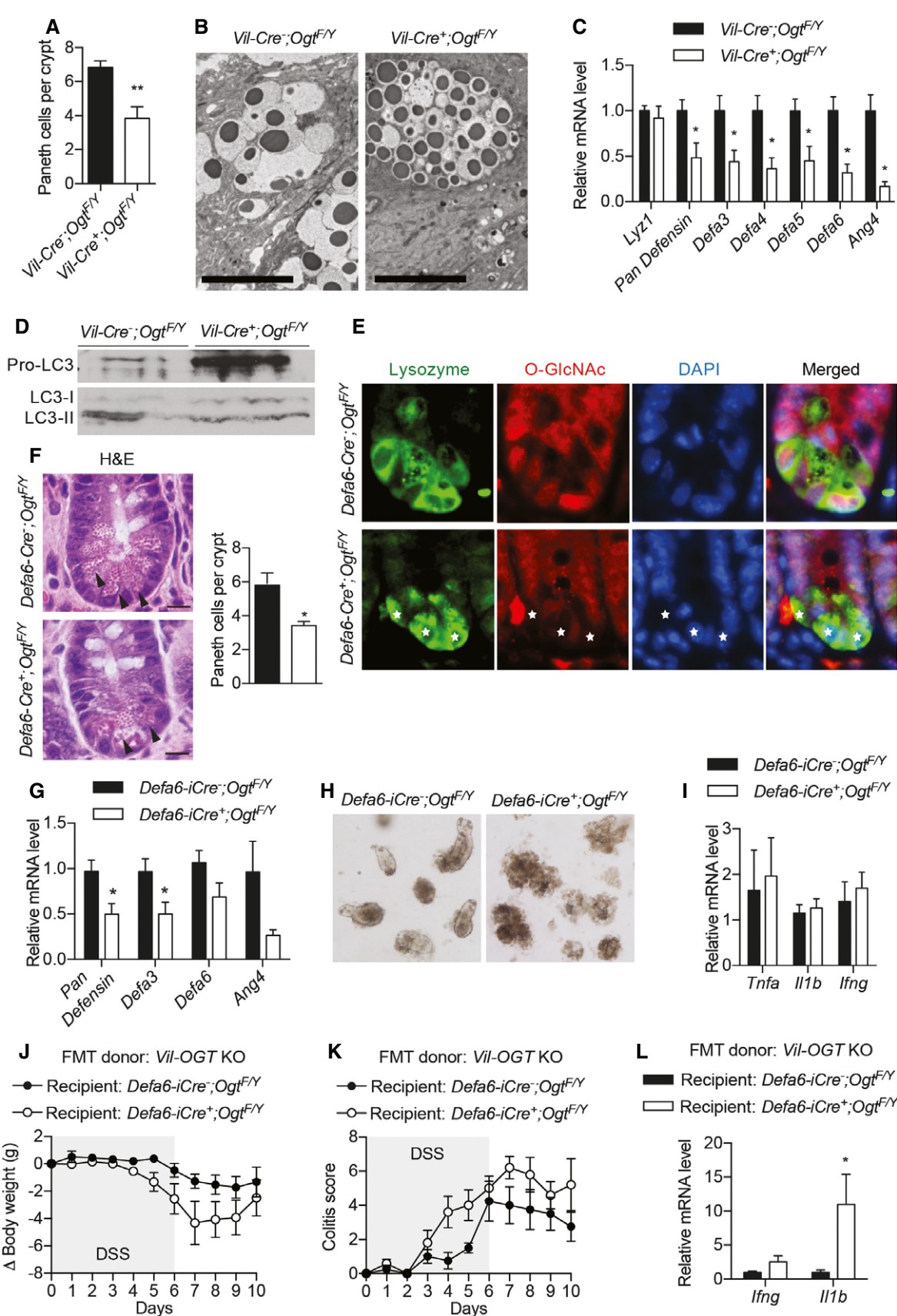

**Figure 6.**

**Figure 6.  Paneth cell dysfunction in OGT-deficient mice.**

A    Numbers of Paneth cells in the ileum of wild-type and *Vil-Ogt* KO mice ($n = 6$, $P = 0.0048$).

B    Representative electron microscopic images of Paneth cells. Scale bars = 1 μm.

C    Expression of AMP genes in ileum tissues of wild-type and *Vil-Ogt* KO mice ($n = 6$, *Pan Defesin* $P = 0.031$, *Defa3* $P = 0.049$, *Defa4* $P = 0.013$, *Defa5* $P = 0.023$, *Defa6* $P = 0.034$, *Ang4* $P = 0.031$).

D    Immunoblotting of LC3 in IEC cells isolated from WT and *Vil-Ogt* KO mice.

E    Representative images of O-GlcNAc staining in Paneth cells of male wild-type and *Defa6-Ogt* KO mice. Stars indicate lysozyme-positive Paneth cells.

F    Morphology and numbers of Paneth cells in the ileum of wild-type and *Defa6-Ogt* KO mice ($n = 5$, $P = 0.024$). Arrow indicates Paneth cell. Scale bars = 10 μm.

G    Expression of AMP genes in the ileum of wild-type and *Defa6-Ogt* KO mice ($n = 5$ *Pan Defesin* $P = 0.0291$, *Defa3* $P = 0.0462$).

H    Representative images of cultured organoid on Day 3.

I    Expression of inflammatory markers in the ileum of wild-type and *Defa6-Ogt* KO mice ($n = 5$).

J–L    Antibiotic-treated wild-type and *Defa6-Ogt* KO male mice were transplanted with fecal microbiota from *Vil-Ogt* KO mice and then induced with DSS for colitis. Daily changes in body weight (J), colitis scores (K), and RT–qPCR of inflammatory genes in the ileum (L) are shown ($n = 4$–5). (L: $P = 0.049$).

Data information: Data are represented as mean ± SEM. *$P < 0.05$; **$P < 0.01$ by two-tailed *t*-test.

Source data are available online for this figure.

by immunofluorescent staining using an anti-lysozyme antibody (Fig EV5). The expression of AMP genes was also downregulated (Fig 6G). Paneth cell is an important source of niche factors to support intestinal stem cells in culture (Barker, 2014). We found that the majority of cultured ileal organoids from *Defa6-Ogt* KO mice could not survive (Fig 6H). These data suggest that protein O-GlcNAcylation is indispensable for the survival and function of Paneth cells.

However, we did not observe any intestinal inflammation in *Defa6-Ogt* KO mice (Fig 6I). Intestinal structure and barrier were largely intact in *Defa6-Ogt* KO mice, indicating that Paneth cell dysfunction alone is not sufficient to precipitate intestinal inflammation. Recent evidence suggests that a second "hit" is required for Paneth cell dysfunction to induce intestinal inflammation (Kaser *et al*, 2008). We then asked whether alterations in gut microbiota together with the disruption of Paneth cell function instigate a pro-inflammatory response. Control and *Defa6-Ogt* KO mice were treated with an antibiotic cocktail and then both received FMT from *Vil-Ogt* KO mice. When colitis was induced by DSS, *Defa6-Ogt* KO mice receiving the dysbiotic microbiota derived from *Vil-Ogt* KO mice had a tendency to lose more body weight (Fig 6J), showed more severe colitis (Fig 6K), and expressed higher levels of inflammatory genes including *Ifng* and *Il1b* (Fig 6L), compared to control mice receiving the same FMT. Collectively, these data demonstrate that intestinal epithelial OGT regulates multiple homeostatic modules, including gut microbiota

and Paneth cell function, to prevent from spontaneous and induced intestinal inflammation.

**Defective O-GlcNAcylation of STAT proteins in *Vil-Ogt* KO mice**

We then sought to determine the molecular mechanisms by which OGT controls intestinal homeostasis. Key regulators in intestinal inflammation such as NF-κB and STAT3 can be modified and regulated by O-GlyNAcylation (Yang *et al*, 2015; Li *et al*, 2017); however, we did not observe any changes in the phosphorylation of NF-κB or STAT3 in the colon of *Vil-Ogt* KO mice (Fig 7A). To identify other potential molecules and pathways modulated by O-GlcNA-cylation, we performed RNA sequencing of isolated IECs from the ileum and colon of control and *Vil-Ogt* KO mice. The sample correlation matrix and hierarchical clustering based on the correlation showed that IECs from control and *Vil-Ogt* KO mice clustered separately (Fig 7B). We identified 3,247 and 2,031 genes that significantly show more than twofold changes in expression in ileal and colonic IECs, respectively (Fig 7C). These differentially expressed genes were then subjected for Ingenuity Pathway Analysis (IPA). Many inflammatory pathways were enriched, such as atherosclerosis signaling, IL-10 signaling, Th1 and Th2 pathways, and T-cell receptor signaling (Fig 7D). To find potential upstream regulators, a bioinformatics program, distant regulatory elements of coregulated genes (DiRE; Gotea & Ovcharenko, 2008), was used to predict common regulatory elements for these differentially expressed

**Figure 7.  O-GlcNAc regulation of STAT1 in IECs.**

A    Expression of phosphorylated and total STAT3 and NF-κB p65 in the colon of wild-type and *Vil-Ogt* KO mice.

B    Ileum and colon IECs from 10-week-old wild-type and *Vil-Ogt* KO mice were isolated for RNA-sequencing analyses. The heatmaps of the sample-to-sample distances are shown.

C    Volcano plots of gene expression changes. The *x*-axis specifies the fold changes (FC), and the *y*-axis specifies the negative logarithm to the base 10 of the *P*-values adjusted for multiple testing with the Benjamini–Hochberg procedure. FC > 2 and $P < 0.05$ genes are shown in solid dots.

D    Most significantly enriched canonical signaling pathways in differentially expressed genes, determined by the Comparison Analysis module in the IPA program.

E    Top candidate transcription factors controlling the observed differentially expressed genes, predicted by DiRE.

F    Levels of STAT1 protein in IECs determined by immunoblotting and O-GlcNAc modification of STAT1 determined by immunoprecipitation using the O-GlcNAc antibody followed by Immunoblotting.

G, H    Caco-2 cells were treated with 100 uM Ac$_4$5S-GlcNAc for 24 h, and levels of STAT1 and global O-GlcNAcylation (G) and STAT1 O-GlcNAcylation (H) were determined.

I    siRNAs against scrambled sequence or human OGT were transfected in 293T cells, and STAT1 levels were determined 48 h later.

J    The pGAS-luc Cis-Reporter plasmid was cotransfected with siRNAs in 293T cells for luciferase assay to determine the transcriptional activity of STAT1 ($n = 4$, $P = 0.0003$).

Data information: Data are represented as mean ± SEM. ***$P < 0.001$ by two-way ANOVA followed by Tukey's multiple comparisons test.

Source data are available online for this figure.

genes. STAT proteins including STAT1 were predicted as top candidate transcription factors (Fig 7E). It has been shown that many STAT proteins can be modified and regulated by O-GlcNAcylation (Gewinner *et al*, 2004; Freund *et al*, 2017), and loss of STAT proteins potentiates the intestinal injury and mucosal inflammation (Gilbert *et al*, 2012; Willson *et al*, 2013; Chiriac *et al*, 2017). We found that in KO IECs, levels of total and O-GlcNAcylated STAT1 were reduced (Fig 7F), indicating that O-GlcNAcylation may control the stability of STAT1 proteins (Ruan *et al*, 2013a).

To test the regulation of STAT1 protein levels by O-GlcNAcylation, we first treated Caco-2 cells with an inhibitor of OGT, Ac₄S-GlcNAc (Gloster *et al*, 2011). Global O-GlcNAcylation was dramatically downregulated by Ac₄S-GlcNAc, as well as the levels of STAT1 total protein (Fig 7G) and STAT1 O-GlcNAcylation (Fig 7H). In 293T cells, knocking down OGT also reduced the levels of STAT1 protein (Fig 7I). More importantly, OGT knockdown significantly blunted the transcription activity of STAT1 (Fig 7J). These data suggest that defective STAT1 signaling

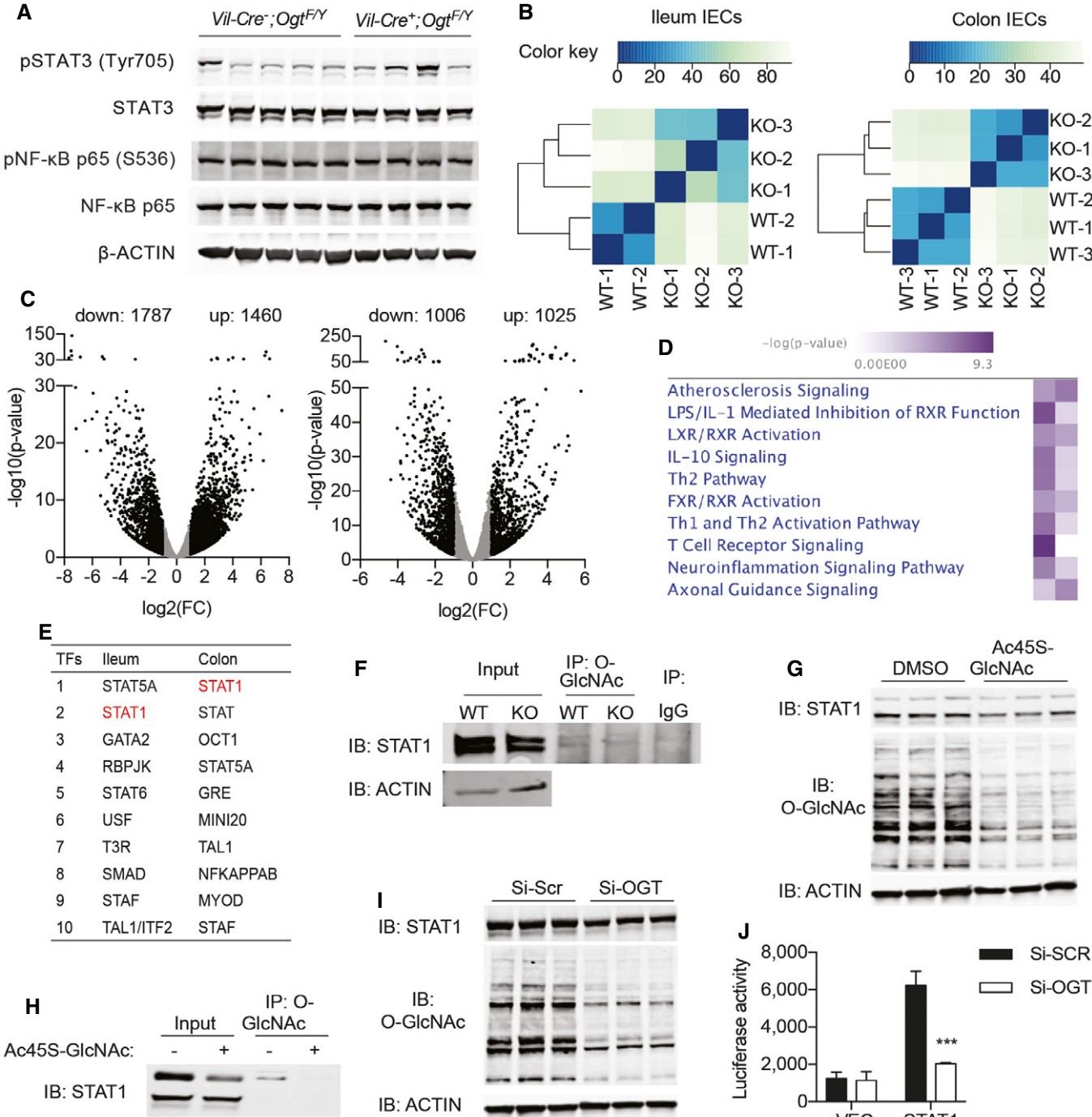

**Figure 7.**

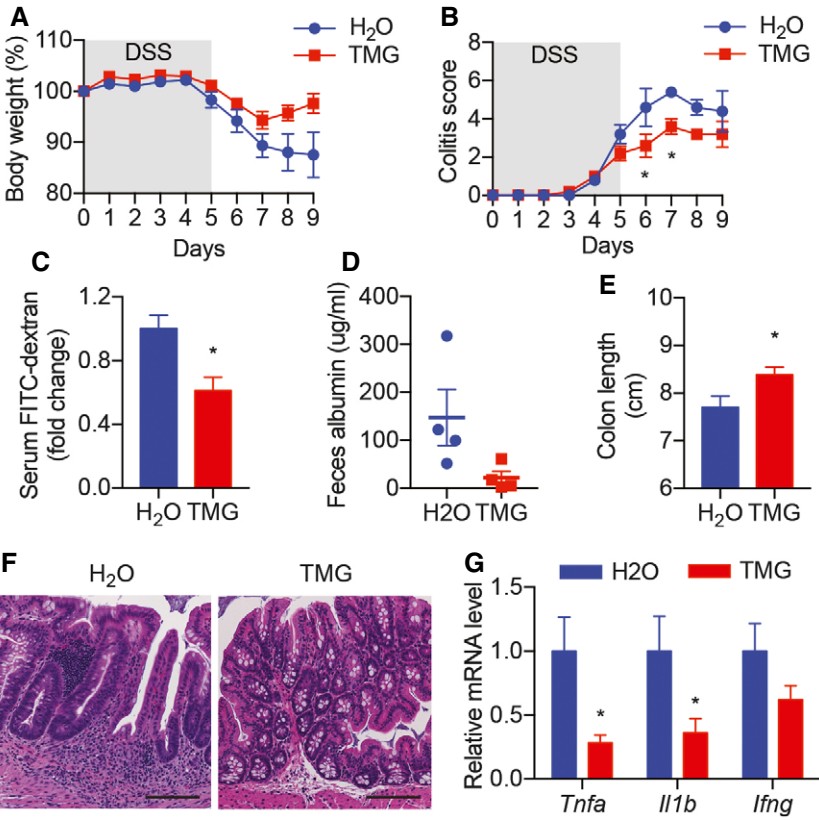

**Figure 8.  Elevation of protein O-GlcNAcylation diminishes DSS-induced intestinal inflammation.**

A, B    Wild-type mice were treated with H2O or TMG for 2 weeks before the induction of colitis by DSS. Daily changes in body weight (A) and colitis score (B; $n = 5$).
(B: Day 6 $P = 0.0171$, Day 7 $P = 0.0445$).

C    Intestinal barrier functional assays measuring serum FITC-dextran before DSS treatment ($n = 5$, $P = 0.0117$).

D–G    Albumin levels in feces (D), colon length (E), H&E staining of colon tissues (F), and RT–qPCR of inflammatory genes in the colon (G) on Day 4 of recovery ($n = 5$).
Scale bars = 100 μm. (E: $P = 0.0474$; G: Tnfa $P = 0.0190$, Il1b $P = 0.0301$).

Data information: Data are represented as mean ± SEM. *$P < 0.05$ by two-way ANOVA followed by Bonferroni corrections (B) and two-tailed $t$-test (C, E, and G).

caused by the loss of O-GlcNAcylation may contribute to intestinal defects observed in *Vil-Ogt* KO mice.

### Inhibition of OGA diminished DSS-induced intestinal inflammation

We have shown that O-GlcNAc signaling is defective in IBD patients and *Ogt* knockout in IECs leads to intestinal damage and inflammation in mice. To determine whether pharmacologically elevated protein O-GlcNAcylation strengthens barrier function and protects mice from chemical-induced acute colitis, we orally administered water or Thiamet-G (TMG, an OGA inhibitor) to C57BL/6 mice ($n = 5$) for 2 weeks. The mice were then treated with 2.5% DSS in drinking water for 5 days followed by 4 days of water only. TMG-treated mice lost less body weight (Fig 8A) and had lower colitis score (Fig 8B) measured as described in Alenghat *et al* (2013). *In vivo* barrier functional assay showed that TMG treatment reduced intestinal permeability before and after DSS induction (Fig 8C and D). TMG-treated mice also had longer colon length (Fig 8E), accelerated mucosal recovery (Fig 8F), and lower levels of inflammatory genes such as *Tnfa* and *Il1b* (Fig 8G). These data demonstrate that augmenting global

protein O-GlcNAcylation by TMG pretreatment ameliorates chemical-induced acute colitis.

## Discussion

Genomewide association studies (GWASs) have identified more than 160 IBD-associated gene loci (Khor *et al*, 2011; Cleynen & Vermeire, 2015); however, the heterogeneity of IBD and the low disease penetrance in individuals carrying disease-susceptibility alleles suggest that complex environment–host interactions cause IBD (Kayama & Takeda, 2012; Kamada *et al*, 2013; Knights *et al*, 2013; Sonnenberg & Artis, 2015). A multi-hit model of IBD has thus been proposed (Maloy & Powrie, 2011). The induction and perpetuation of intestinal inflammation require the convergence of several abnormalities that affect overlapping layers of regulatory modules, including genetic susceptibility, barrier defects, microbial dysbiosis, and sustained innate immunity. Humans or animals with defects in one layer of these modules are only predisposed to but not sufficient to develop IBD.

Here, we report that protein post-translational O-GlcNAc modification in the intestinal epithelium is compromised in human IBD.

IEC-specific knockout of OGT leads to intestinal damage and inflammation in mice. The early-onset spontaneous intestinal damages observed in *Vil-Ogt* KO mice suggests that protein O-GlcNAcylation is a regulator of multiple homeostatic modules in the epithelium. We demonstrated that O-GlcNAc deficiency results in disruptive epithelial barrier, Paneth cell dysfunction, and microbial dysbiosis. Inducing Paneth cell disorder by knocking out the *Ogt* gene specifically only in *Defa6*-expressing cells or transplanting fecal microbiota from *Vil-Ogt* KO mice into antibiotic-treated wild-type mice did not cause intestinal inflammation, again supporting the multi-hit model of IBD (Maloy & Powrie, 2011). Nevertheless, microbial dysbiosis and Paneth cell dysfunction together may potentiate chemical-induced inflammation.

Many genetically engineered mouse models have been established to understand the molecular mechanisms and to develop therapeutic strategies for IBD (Mizoguchi, 2012; Kiesler *et al*, 2015). However, only a handful of IEC-specific genetic models were shown to develop spontaneous intestinal inflammation; these include *NEMO^{IEC-KO}*, *IKK1/2^{IEC-KO}*, *TAK1^{IEC-KO}*, *XBP1^{IEC-KO}*, and *ATG16L1^{IEC-KO}* mice (Nenci *et al*, 2007; Kajino-Sakamoto *et al*, 2008; Kaser *et al*, 2008; Tschurtschenthaler *et al*, 2017). NEMO, IKK1 and 2, and TAK1 are all required for the activation of nuclear factor kappa B (NF-κB), suggesting that IEC-specific NF-κB signaling is protective in intestinal inflammation. Abnormal autophagy and unresolved endoplasmic reticulum (ER) stress are common features of IBD (Cadwell *et al*, 2008; Kaser *et al*, 2008). Dysfunctions in either unfolded protein response factor, X-box binding protein 1 (XBP1), or autophagy (ATG16L1 or ATG7) in IECs result in the reciprocal compensatory, and severe spontaneous ileitis develops if both mechanisms are defective (Adolph *et al*, 2013).

O-GlcNAcylation has been associated with both pro- and anti-inflammatory effects in various conditions (Baudoin & Issad, 2014; Zheng *et al*, 2017; Thi Do *et al*, 2018). The NF-κB complex is modified by O-GlcNAc; however, O-GlcNAcylation has both positive (James *et al*, 2002; Yang *et al*, 2008; Allison *et al*, 2012; Ramakrishnan *et al*, 2013; Zhang *et al*, 2015) and negative (Zou *et al*, 2009; Xing *et al*, 2011) effects on NF-κB activity and inflammation based on reported results. Here, we did not observe any changes in NF-κB activity in *Vil-Ogt* KO mice, suggesting intestinal epithelial O-GlcNAcylation is not a major driver of the NF-κB pathway. In contrast to our findings, a recent work showed that OGA^{+/−} knockout mice are susceptible to DSS-induced colitis, suggesting that hyper-O-GlcNAcylation may promote intestinal inflammation by activating NF-κB signaling (Yang *et al*, 2015). However, in the whole-body heterozygous knockout mouse model used by Yang *et al*, it is unclear what cell types are important in mediating the function of OGA in DSS-induced colitis. Given the protective role of NF-κB in IEC (Nenci *et al*, 2007; Kajino-Sakamoto *et al*, 2008), it is unlikely that the elevation of protein O-GlcNAcylation in IECs contributes to the phenotype. In addition, Li *et al* (2017) showed that inhibition of OGT in macrophage by myeloid-derived cullin 3 protects against intestinal inflammation. They found that the E3 ubiquitin ligase cullin 3 downregulated the expression of *OGT* gene, thereby inhibiting STAT3 O-GlcNAcylation. Since both NF-κB and STAT3 are critical components in immune activation, we propose that protein hyper-O-GlcNAcylation in the immune system may promote immune activation and intestinal inflammation.

On the other hand, both NF-κB and STAT signaling pathways are indispensable for intestinal epithelial homeostasis, as IEC-specific deficiencies in either the NF-κB complex or STAT family proteins cause or predispose to intestinal inflammation (Nenci *et al*, 2007; Kajino-Sakamoto *et al*, 2008; Gilbert *et al*, 2012; Willson *et al*, 2013; Chiriac *et al*, 2017). By RNA sequencing of IECs, we predicted STAT proteins, particularly STAT1, as a downstream target of protein O-GlcNAcylation and mediator of intestinal damage in *Vil-Ogt* KO mice. Several STAT proteins including STAT1, STAT3, STAT5A, STAT5B, and STAT6 can be O-GlcNAc-modified (Gewinner *et al*, 2004; Freund *et al*, 2017). We confirmed that STAT1 in IECs was O-GlcNAcylated, and loss of OGT reduced STAT1 expression and transcription activity. In the intestinal epithelium, deficiencies in STAT1, STAT3, or STAT5A did not lead to spontaneous inflammation, even though these mice were prone to chemical-induced colitis (Gilbert *et al*, 2012; Willson *et al*, 2013; Chiriac *et al*, 2017). Redundancy and compensation between different STAT proteins may attenuate the detrimental effects caused by the loss of individual STAT proteins. However, in *Vil-Ogt* KO mice, impairment in O-GlcNAc signaling is likely to dysregulate multiple STAT proteins to elicit profound inflammation in the epithelium. Further studies on the O-GlcNAcylation of specific STAT proteins are warranted to delineate the precise control and function of protein O-GlcNAcylation in intestinal homeostasis and inflammation.

In human IBD, levels of OGT protein expression and global O-GlcNAcylation in epithelial cells are greatly decreased. Linear regression analysis indicated an inverse correlation between disease severity and OGT/O-GlcNAc levels. Statistical significance was reached even with limited numbers of samples, emphasizing the clinical importance of epithelial O-GlcNAcylation in intestinal pathology. However, we could not observe such correlations within disease-only groups (data not shown), probably due to the small sample size and the lack of a broad spectrum of disease severity. A larger cohort, with controlled age, gender, diet, disease status, and treatment, is needed in the future to elucidate the relationship between the reduction in OGT and the progress of disease. Nevertheless, no mutations in or polymorphisms associated with the *OGT* gene have be reported, probably because the *OGT* gene is resided in the chromosome X, which is commonly excluded from GWAS analyses. Further investigations are required to determine the cause of deficiencies in protein O-GlcNAcylation in IBD.

Currently, there is no cure for IBD. Patients rely on diet and lifestyle changes, drug therapy, or surgery to relieve symptoms and induce remission. A remarkable observation of this study is that pretreatment with an OGA inhibitor to elevate the intestinal epithelial O-GlcNAc modification protects mice from chemical-induced colitis. It is warranted to test whether genetic and chemical approaches to increase O-GlcNAc levels also have therapeutic effects on chronic intestinal inflammation. Our study will shed light on the future design of novel preventions and therapeutics for IBD.

# Materials and Methods

### Human samples

De-identified intestine slides from UC and CD patients were purchased from the Biological Materials Procurement Network

(BioNet) at the University of Minnesota. Normal colon tissues adjacent to tumor from age-matched subjects were used as controls. The Institutional Review Board of the University of Minnesota has reviewed our study and determined that it did not meet the regulatory definition of human subject research. Informed consent was obtained from all Chinese subjects enrolled in the study for biopsy sampling per protocol approved by the Institutional Review Board at the Fourth Military Medical University. The intensity of OGT and O-GlcNAc staining was semi-quantified by visually scoring. 10–20 crypts per sample were randomly selected and ranked 1–4 based on the staining intensity (1 being weakest and 4 being strongest). Average scores were used for comparison between control, UC, and CD patients. For the correlation between the disease severity and the level of O-GlcNAcylation, slides from another cohort of 10 controls, 10 UCs, and 9 CDs were obtained from BioNet (Appendix Table S2). Tissue inflammation and damage were evaluated by the Geboes scoring system for UC and the global histological activity (GHA) scoring system for CD (Dataset EV1; Levesque et al, 2015). Lineage regression analysis was performed to determine the correlation.

## Animal

*Ogt-floxed* mice on the C57BL/6 background (Shafi et al, 2000) were kindly provided by Dr. Xiaoyong Yang (Yale University). *Villin-Cre* mice (stock no. 004586) were purchased from the Jackson Laboratory. *Defa6-iCre* mice on the C57BL/6 background were generated and reported earlier (Adolph et al, 2013). Homozygous floxed $Ogt^{F/F}$ mice were bred to *Villin-Cre* or *Defa6-iCre* mice to generate *Vil-Ogt* KO and *Defa6-Ogt* KO mice, respectively. 10- to 15-week-old male mice were used for experiments unless otherwise mentioned. All animals were kept on a 14-h: 10-h light: dark cycle in the animal facility at the University of Minnesota or on a 12-h: 12-h light: dark cycle at Yale University. Mice were group-housed unless otherwise mentioned, with free access to water and standard chow diet. All procedures involving animals were conducted within IACUC guidelines under approved protocols.

## In vivo intestinal barrier function assays

Mice with free access to water were fasted for 2-h and orally gavaged with fluorescein isothiocyanate (FITC)-dextran (average molecular weight: 3,000–5,000, 0.6 mg/g; Sigma) diluted in PBS. Fluorescence intensity of plasma samples was measured (excitation 492 nm/emission 520 nm) 4 h after the gavage. For fecal albumin assays, fecal pellets were weighed and homogenized in diluent (PBS, 1% BSA, 0.05% Tween 20). Albumin levels in fecal homogenates were measured by ELISA (Bethyl Laboratories) according to the manufacturer's protocol.

## Fecal microbiota colonization

Donor mice were euthanized, and ceca were aseptically removed immediately. The content was diluted 1:10 in a 50% glycerol/PBS solution and frozen at −80°C. On the day of inoculation, the cecal content was further diluted 1:5 in autoclaved PBS prior to oral gavage of 0.15 ml per mouse. Prior to inoculation, recipient animals were treated with a cocktail of broad-spectrum antibiotics to deplete gut microbiota (Hill et al, 2010; Reikvam et al, 2011). Specifically,

animals having free access to autoclaved food and water were subjected to oral gavage daily for 14 days with 100 μl of autoclaved water supplemented with ampicillin (2 mg/ml), gentamicin (2 mg/ml), metronidazole (2 mg/ml), neomycin (2 mg/ml), and vancomycin (1 mg/ml).

## Murine colitis model

36–50 kDa colitis-grade dextran sodium sulfate (DSS; MP Biomedicals) was added to drinking water at 2.0 or 2.5% weight/volume for 5–6 days and then removed for the recovery. Thiamet-G (TMG; CarboSynth) at a dose that sufficiently increases O-GlcNAc level (0.2 g/kg BW; Yuzwa et al, 2008) was gavaged daily from 2 weeks before the DSS treatment until the end of experiment. Disease was scored as described in Alenghat et al (2013): (i) weight loss (no loss = 0; < 5% = 1; 5–10% = 2; 10–20% = 3; > 20% = 4); (ii) stool (normal = 0; soft, watery =1; very soft, semi-formed = 2; liquid, sticky, or unable to defecate = 3); (iii) blood (no blood = 0; visible blood in rectum = 1; visible blood on fur = 2); and (iv) general appearance (normal = 0; piloerection = 1; lethargy and piloerection = 2; motionless = 4). Histological injury and inflammation were scored as described in Gilbert et al (2012). Scoring parameters included edema (scale: 1–4), erosion/ulceration of the epithelial monolayer (scale: 1–4), crypt loss/damage (scale: 1–4), and infiltration of immune cells into the mucosa (scale: 1–4).

## Colonoscopy

Mice were anaesthetized using intraperitoneal injection of ketamine, xylazine, and acepromazine. The "Coloview system" (Karl Storz) was used for colonoscopy as described in Becker et al (2005). The endoscopic procedure was viewed on a color monitor, and pictures were digitally recorded.

## Ileal organoid and Caco-2 cell culture

Isolated ileal crypts were counted and cultured in Matrigel following a published protocol (Mahe et al, 2013). Advanced DMEM/F12 was supplemented by EGF (50 ng/ml), Noggin (100 ng/ml), and R-spondin conditional media. Caco-2 cells were cultured as described previously (Natoli et al, 2012). Differentiated Caco-2 cells were treated with 100 μM Ac$_4$5S-GlcNAc for 24 h and subjected to protein extraction.

## Antibodies

Anti-O-GlcNAc (RL2, ab2739) was from Abcam. Anti-ZO-1 (617300) was from Life technologies. Anti-ZO-2 (18900-1-AP), anti-CLAUDIN 3 (16456), and anti-OCCLUDIN (13409) were from proteintech. Anti-Ki67 (550609) was from BD Biosciences. Anti-Myeloperoxidase (MPO, PA5-16672) was from Thermo Fisher. Anti-F4/80 (123102) and Anti-CD4 (100506) were from Biolegend. Anti-OGT (#24083), anti-Cleaved CASPASE 3 (#9661), anti-LC3B (#2775), anti-α-E-CATENIN (#3042), anti-β-CATENIN (#8480), anti-PLAKOGLOBIN (#2309), anti-STAT1 (#9172), anti-STAT3 (#4904), anti-phospho-STAT3 (Tyr705, #9145), anti-STAT5 (#9363), anti-NF-κB p65 (#8242), and anti-phospho-NF-κB p65 (Ser536, #3033) were from Cell Signaling Technology. Anti-lysozyme (CSB-PA02769A0Rb) was

from Cusabio. Anti-DESMOPLAKIN I/II (sc-33555) was from Santa Cruz Biotechnology. Anti-β-ACTIN (A5441) and anti-CLAUDIN 5 (ABT45) were from Millipore Sigma.

## Histology, immunohistochemistry, immunofluorescence, and pathological analysis

Tissues were fixed in 10% neutral buffered formalin. Paraffin sections of intestine tissues were stained with hematoxylin and eosin (H&E) and alcian blue (Sigma) according to standard procedures. Immunohistochemistry was carried out using Histostain-Plus 3rd Gen IHC Detection Kit (Life technologies) following manufacturer's instruction. Antigen retrieval was performed in citric buffer using a 2100 Retriever (Aptum Biologics). For immunofluorescence, tissue slides were blocked with 3% BSA, 0.2% TWEEN 20 in PBS, incubated with primary antibodies (1:100 dilution) overnight, and secondary antibodies (Alexa Fluor 488 anti-Rabbit IgG, Alexa Fluor 594 anti-Rabbit IgG, and Alexa Fluor 594 anti-Mouse IgG, 1:400) for 1 h. TUNEL assay was carried out using *In Situ* Cell Death Detection Kit (Roche), following manufacturer's instruction. A Nikon system was used for fluorescence detection. A modified histopathologic scoring system was used to analyze pathological alterations in the intestinal mucosa (Gilbert *et al*, 2015). Briefly, scores of epithelial hyperplasia (scale: 1–4), epithelial hypertrophy (scale: 1–4), crypt elongation (scale: 1–4), villus length (scale: 1–4), cell apoptosis (scale: 1–4), immune cell infiltration (scale: 1–4), and mucosal thickness (scale: 1–4) were combined.

## Immunoprecipitation and Western blot

Tissues were lysed in RIPA or NP-40 buffer containing proteinase inhibitors, protein phosphatase inhibitors, and an OGA inhibitor. For immunoprecipitation, whole-cell lysates were mixed with various antibodies as specified in text and precipitated by Protein A/G agarose beads (Santa Cruz). Equal amounts of whole lysates or immunoprecipitation samples were electrophoresed on TGX precast gels (Bio-Rad) and transferred to nitrocellulose membrane. Primary antibodies (1:500 dilution for antibody from Santa Cruz Biotechnology and 1–1,000 dilution for the rest) were incubated at 4°C for overnight. Western blotting was visualized by using IRDye secondary antibodies and the Odyssey imaging system (LI-COR Biosciences).

## RNA and real-time PCR

Total RNA was extracted from mouse tissues and cells using TRIzol reagent (Invitrogen). cDNA was reverse-transcribed (Bio-Rad) and amplified with SYBR Green Supermix (Bio-Rad) using a C1000 Thermal Cycler (Bio-Rad). All data were normalized to the expression of the Rplp0 gene. Primer sequences are as follows: *Ang4* for 5′-TCTCCAGGAGCACACAGCTA-3′; *Ang4* rev 5′-ACAACAAAGGAC ATGGGCTC-3′; *Defa3* for 5′-GTCCAGGCTGATCCTATCCA-3′; *Defa3* rev 5′-AGAGCCTTCTGGGTCTCCA-3′; *Defa4* for 5′-GTCCAGGCT GATCCTATCCA-3′; *Defa4* rev 5′-TGGCCTCCAAAGGAGATAGA-3′; *Defa5* for 5′-TCCAGGCTGATCCTATCCAC-3′; *Defa5* rev 5′-TGGCC TCCAAAGGAGATAGA-3′; *Defa6* for 5′-GGACCAGGCTGTGTCTGTC T-3′; *Defa6* rev 5′-TTGCAGCCTCTTGCTCTACA-3′; *Ifng* for 5′-TCAA GTGGCATAGATGTGGAAGAA-3′; *Ifng* rev 5′-TGGCTCTGCAGGATT

TTCATG-3′; *Il1b* for 5′-TGTGAAATGCCACCTTTTGA-3′; *Il1b* rev 5′-GGTCAAAGGTTTGGAAGCAG-3′; *Il6* for 5′-GAGGATACCACTCC CAACAGACC-3′; *Il6* rev 5′-AAGTGCATCATCGTTGTTCATACA-3′; *Lyz1* for 5′-CTGTGGGATCAATTGCAGTG-3′; *Lyz1* rev 5′-GAATGCCT TGGGGATCTCTC-3′; *Ogt* (human) for 5′-CTGTCACCCTTGACC CAAAC-3′; *Ogt* (human) rev 5′-CTCTGGGAAGACTTCTAATGC-3′; *Pan Defensin* for 5′-GGTGATCATCAGACCCCAGCATCAGT-3′; *Pan Defensin* rev 5′-AAGAGACTAAAACTGAGGAGCAGC-3′; *Rplp0* for 5′-AGATGCAGCAGATCCGCAT-3′; *Rplp0* rev 5′-GTTCTTGCCCAT CAGCACC-3′; *Tnfa* for 5′-CATCTTCTCAAAATTCGAGTGACAA-3′; *Tnfa* rev 5′-TGGGAGTAGACAAGGTACAACCC-3′.

## Electron microscopy

Mice were perfused with PBS followed by 4% PFA. Intestinal tissues were cut into less than $1 \times 1 \times 1$ mm$^3$ cubes for post-fixation overnight in 2.5% gluteraldehyde and 2% PFA in 0.1 M sodium cacodylate buffer. Embedding, sectioning, and observation were carried out at the Electron Microscopy Core at Yale School of Medicine.

## 16S rRNA gene sequencing

Total DNA in stool and cecal contents was extracted using the PowerFecal DNA Isolation Kit (MoBio). The V4 region of the bacterial 16S rRNA gene was amplified by triplicate PCR (F515/R806) using barcoded fusion primers. Samples were pooled in sets with a maximum of 96 samples in equal quantities. Paired-end sequencing of the amplicon library was performed on the Illumina MiSeq 300-bp paired-end platform at the University of Minnesota Genomics Center (Gohl *et al*, 2016). A multi-step bioinformatics analysis was performed using the QIIME 1.9.1 software, including filtering raw fastq files for primer and adapter dimer sequences, removing contaminating host sequences and chimeric sequences, clustering sequences into operational taxonomic units (OTUs) using the open-reference OTU calling method with the greengenes 16S reference, and calculating alpha and beta diversity metrics. Linear discriminant analysis (LDA) effect size (LEfSe) method was used for microbial biomarker discovery (Segata *et al*, 2011).

## RNA sequencing

IECs from ileum and colon were isolated by shaking intestinal tissue in PBS with 10 mM EDTA at 37°C for 30 min. Total RNA was extracted using the RNeasy Plus Mini Kit (Qiagen) following the manufacturer's instruction. RNA concentration and integrity of each sample were measured on an Agilent Bioanalyzer. Equal amounts of RNA from each sample were pooled for cDNA library construction and sequencing on an Illumina HiSeq instrument with $1 \times 75$ bp reads at the Yale Center for Genome Analysis. Sequencing analysis was performed on the Galaxy server hosted by the Minnesota Supercomputing Institute. FastQ files were trimmed using Trimmomatic. Quality control checks on raw sequence data for each sample were performed with FastQC. Read mapping was performed via TopHat using the UCSC mouse genome (mm 10) as reference. DESeq2 was used for differential gene expression analysis. The Ingenuity Pathway Analysis (IPA) tool from Qiagen was used for pathway analysis and upstream regulator analysis of significant genes that showed more than twofold of differential expression. Finally, these

coregulated genes were subjected to the Distant Regulatory Elements (DiRE) server to predict common regulatory elements (Gotea & Ovcharenko, 2008).

## Luciferase assay

HEK 293T cells (ATCC) were seeded at 48-well plates while scramble and OGT siRNA was transfected using Lipofectamine 2000 (Invitrogen). On the second day, cells were transfected with pGAS-Luc (IFN-γ-activated sequence) plasmid (Agilent) with pcDNA or STAT1α plasmid (Addgene). The reporter activity was measured 36 h after pGAS-Luc transfection with the Dual-Luciferase® Reporter Assay System (Promega) following manufacturer's instruction.

## Statistical analyses

Mice receiving FMT or drug treatment were randomly assigned to control or treatment groups. The histological scores were provided by a professional pathologist without any information of the treatment group. Otherwise, no blinding was done, due to the obvious size difference between WT and KO mice for most measurements. Results are shown as mean ± SEM. Shapiro–Wilk normality test was used to assess normal distribution. The comparisons were carried out using two-tailed unpaired Student's t-test, one-way ANOVA with the Dunnett post hoc test, and two-way ANOVA followed by post hoc comparisons using Tukey or Bonferroni corrections. The prevalence of rectal prolapse was compared between WT and Vil-Ogt KO mice using the Log-rank (Mantel–Cox) test in Prism.

## Data availability

RNA-Seq data: Gene Expression Omnibus GSE100473.

**Expanded View** for this article is available online.

## Acknowledgements

We thank Dr. Xiaoyong Yang for providing the *Ogt-floxed* mouse line, Dr. David Vocadlo for providing Ac₄5S-GlcNAc, and Dr. Daniel Vallera for providing Caco-2 cells. We thank Dr. Tim Starr for providing the Karl Storz Coloview system and Patrick Blaney for assisting with the colonoscopy. We thank Dr. Dan Knights and Trevor Gould for assisting the analysis of 16S sequencing data. We thank Dr. Weiqi He and Huashan Li for providing the technical assistance in organoid culture. This work was supported by National Natural and Science Foundation of China (81770543), American Heart Association Scientist Development Grant (14SDG20120052), Mizutani Foundation for Glycoscience Grant (170133), and University of Minnesota Medical School 2017 Innovation Grant to H.-B.R, Key Science and Technology Project of Henan Province (182102310107) to X. X, Natural Science Foundation of Jiangsu Province (BK20150687) to Z. H, NIH DK088199 to R.S.B, and Crohn's Colitis Foundation of America Senior Research Award (426234) to X.H.

## Author contributions

MZ, with the help from XX, KR, MC, CS, and ZH, designed, performed, and analyzed most experiments. BX, KW, and YN performed experiments using Chinese human biopsies. XH performed pathological analyses. RSB provided the *Defa6-iCre* mouse strain. H-BR conceived, designed, and performed experiments. MZ and H-BR with contributions from XH and RSB wrote the manuscript.

## The paper explained

### Problem

It is generally accepted that IBD requires the convergence of several abnormalities that affect overlapping layers of regulatory modules including genetic mutations, permeable barriers, bacterial alterations, and immune over-activation. However, we still do not fully understand genes and pathways that contribute to such multi-layer defects.

### Results

Here, we report a post-translational modification on proteins, termed O-GlcNAcylation, in the gut epithelium controls multiple regulatory mechanisms including epithelial barrier, autophagy, and gut bacteria to maintain intestinal homeostasis. We found that levels of protein O-GlcNAcylation were reduced in both UC and CD patients and negatively correlated with disease severity. In mouse models, the deficiency of O-GlcNAcylation in intestinal epithelial cells broke down the gut barrier, disrupted Paneth cell function, and caused changes in microbial composition to eventually elicit intestinal inflammation. On the other hand, a drug that increases O-GlcNAcylation alleviated chemical-induced colitis in mice.

### Impact

Our data reveal that protein O-GlcNAcylation controls multiple layers of regulatory mechanisms in intestinal epithelial cells. Drugs specifically enhancing O-GlcNAcylation in the gut epithelium may prevent and treat IBD.

## Conflict of interest

The authors declare that they have no conflict of interest.

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
