## [Review Process File · EMBO Molecular Medicine]

Deficiency in Intestinal Epithelial O-GlcNAcylation Predisposes to Gut Inflammation

Ming Zhao, Xiwen Xiong, Kaiqun Ren, Bing Xu, Meng Cheng, Chinmayi Sahu, Kaichun Wu, Yongzhan Nie, Zan Huang, Richard S. Blumberg, Xiaonan Han, Hai-Bin Ruan

Review timeline:

Submission date:	06 December 2017
Editorial Decision:	18 February 2018
Revision received:	15 May 2018
Editorial Decision:	23 May 2018
Revision received:	01 June 2018
Accepted:	05 June 2018

Editor: Céline Carret

Transaction Report:

1st Editorial Decision

18 February 2018

Thank you for the submission of your manuscript to EMBO Molecular Medicine. We sincerely apologise for the delay in getting back to you, due to unfortunate staff shortage and travelling. We have now heard back from the two referees we asked to evaluate your manuscript.

As you will see from the set of comments below, the referees find merits in the study. However, they both recommend increasing novelty and conclusiveness, and do provide detailed suggestions to reach this goal. Please note referee 2 concerns about the clinical relevance; as a journal of translational value, we would like to strongly encourage you to provide more clinical insights through patients' data. We also would encourage you to address the limited mechanistic insights and revise the paper as suggested to improve conclusiveness, interest and clinical relevance.

We would therefore welcome the submission of a revised version within three months for further consideration and would like to encourage you to address all the criticisms raised as suggested to improve conclusiveness and clarity. Please note that EMBO Molecular Medicine strongly supports a single round of revision and that, as acceptance or rejection of the manuscript will depend on another round of review, your responses should be as complete as possible.

I look forward to receiving your revised manuscript.

***** Reviewer's comments *****

Referee #1 (Comments on Novelty/Model System for Author):

The authors combine both human data (male/female) and mice models (male/female): in mice, the authors use very elegant approaches of OGT cell-specific (both IEC and paneth cells) deletions and combine a large variety of analyses such as immunohistochemistry together with gut microbiota analysis, gut microbiota transfer, western blots, gene expression analysis and clinical score, etc.

Referee #1 (Remarks for Author):

The work from Zhao and colleagues addresses the role of O-linked β -N-acetylglucosamine (O-GlcNAc) modification and related levels of protein O-GlcNAcylation together with the expression of both O-GlcNAc transferase (OGT) and O-GlcNAcase (OGA), the enzyme removing O-GlcNAcylation, on intestinal homeostasis and, in particular, inflammation in IBD (UC and CD). The authors combine both human data (male/female) and mice models (male/female): in mice, the authors use very elegant approaches of OGT cell-specific (both IEC and paneth cells) deletions and combine a large variety of analyses such as immunohistochemistry together with gut microbiota analysis, gut microbiota transfer, western blots, gene expression analysis and clinical score, etc. Data provided on humans are quite clear but a few questions arise in the mouse model, as reported below:

- Please show the knockout specificity and efficiency in Vil-OGT KO female mice and in defa6-OGT KO mice.
- Fig.2F is difficult to read and no major changes appear to be there according to pictures provided, except for colon. Please apply a mucosal injury score to quantify what shown.
- What about the relative mRNA level of Defa6?
- The interest in Plakoglobin is not clear. Proteins of tight junctions better explain the differences in permeability. Please provide WB analysis for some claudins (at least 3 and 5) and occludin.
- Data on Fig.5D, Fig.6H,I Fig.8B are over-interpreted and must not be analysed by t-test, due to the kinetic nature of the analyses, in which points are connected to each other. A 2-ANOVA with a Bonferroni (or Holm-Sidak, more power) post-hoc test must be applied instead. According to the new results the underlined sentence "Using fecal microbiota transplantation and Paneth cell-specific deletion of OGT in mice, we demonstrate that microbial dysbiosis and Paneth cell dysfunction are insufficient to induce spontaneous inflammation, but exacerbate chemical-induced colitis" may need to be revised.
- Page 12 of the merged pdf: explain the rationale to use only 2-hour fasting for FITC-dextran; please specify whether water was removed or not; the molecular weight of the dextran used is missed.
- Page 12 of the merged pdf: in the "murine colitis model" chapter the right symbol for molecular weight is kDa (and not kD)

Referee #2 (Comments on Novelty/Model System for Author):

The authors took advantage of a cell specific knock-out mouse model of OGT in IECs and demonstrated that O-GlcNAcylation is essential for microbial composition, epithelial barrier function and survival of paneth cells. I think this is an adequate experimental model to test the function of this post-translational modification in intestinal epithelium. Moreover, they also used epithelial organoids, and confirmed the decreased survival upon inhibition of O-GlcNAcylation, which indicates the epithelial-intrinsic effect.

Although the link between paneth cell function and inhibition of O-GlcNAcylation was still not known, several previous publications demonstrated the association between this PTM and inflammation. I would suggest a more profound study in terms of paneth cell function in order to emphasize the novelty of the findings.

The results presented in the paper are limited in terms of medical impact so far. However, the confirmation of the hypothesis would lead to future translational perspectives with potential benefit for IBD patients. A correlation between expression data (Fig. 1) and clinical outcome of the patients would be useful in order to provide some more precise clinical impact of the here presented findings.

As mentioned above, I think the model is adequate.

Referee #2 (Remarks for Author):

The manuscript clearly shows that O-GlcNAcylation in intestinal epithelium is crucial for maintenance of epithelial barrier function, microbiota composition and paneth cell function. The latter being the most important finding due to its novelty. Several previous publications have demonstrated the association between O-GlcNAcylation and inflammation in several models and cell types. A reference to these previous publications could be included into the introduction, such as:

Molecules. 2017 Dec 26;23(1).

Clin Exp Immunol. 2017 Dec 16.

Inflamm Res. 2015 Dec;64(12):943-52.

However, this study demonstrate that paneth cell dysfunction might be behind the increased intestinal permeability due to inhibition of O-GlcNAcylation in IECs. I think a more profound study of paneth cells would improve the quality of the manuscript: number of paneth cells in organoids, for instance, activation of autophagy at the crypt bottom.

On the other hand, two publications showed the opposite effect upon modulation of O-GlcNAcylation in myeloid cells or complete knock-out mice.

1. Myeloid-derived cullin 3 promotes STAT3 phosphorylation by inhibiting OGT expression and protects against intestinal inflammation

2. Elevated O-GlcNAcylation promotes colonic inflammation and tumorigenesis by modulating NF- κ B signaling

I agree with the authors that the cell-specific strategy is adequate in order to target a single cell type, as discussed in the manuscript. However, they did not consider the other publication into the discussion, which I think should be mentioned. On the other hand, these two publications demonstrated that STAT3 or NF κ B were the target pathways, respectively. Based on this, I would recommend the authors to provide data in order to discard these two pathways as key players in intestinal epithelium, which would improve the quality of the manuscript.

Concerning the clinical impact of the study, the authors showed the regulation on OGT and OGN expression in intestinal epithelium from IBD patients upon inflammation. Despite this nice link between inflammation and O-GlcNAcylation, a correlation study between clinical and expression data is needed. For instances, it would be nice to show a correlation between the degree of inflammation and the expression of OGT and OGN. Therefore, it would be nice to include more patients into the study.

Minor comments:

Fig. 1: include severity of inflammation from human IBD patients included in the analysis

Fig. 4C: symbols indicating TJs, desmosomes and AJs are not visible.

Fig. 6: paneth cell dysfunction; survival of paneth cells in organoid culture system, for instance

Fig. 7: include STAT3 phosphorylation and NF κ B activation.

1st Revision - authors' response

15 May 2018

Referee #1:

Please show the knockout specificity and efficiency in *Vil*-OGT KO female mice and in *Defa6*-OGT KO mice.

Response: We have now provided the Western Blot results in Fig EV2B showing a dramatic reduction in O-GlcNAcylation levels in the intestine of *Vil-Ogt* KO female mice. The immunofluorescent staining of O-GlcNAcylation in Paneth cells of *Defa6-Ogt* KO mice (new Fig 6E and EV5) showed that OGT was specifically and efficiently knocked out in Paneth cells.

Fig.2F is difficult to read and no major changes appear to be there according to pictures provided, except for colon. Please apply a mucosal injury score to quantify what shown.

Response: Fig 2F shows the general morphology of intestinal tissues of wildtype and *Vil-Ogt* KO mice. High magnification images of ileum and colon are shown in Fig 3A and injury score is determined and shown in Fig 3B.

What about the relative mRNA level of *Defa6*?

Response: As shown in the updated Fig 6C, the mRNA level of *Defa6* was significantly downregulated in the *Vil-Ogt* KO ileum, similar to other defensins. In *Defa6-Ogt* KO mice, the *Defa6* expression also showed a decreasing trend (Fig. 6G).

The interest in Plakoglobin is not clear. Proteins of tight junctions better explain the differences in permeability. Please provide WB analysis for some claudins (at least 3 and 5) and occludin.

Response: The *Vil-Ogt* KO mice had impaired barrier function and disrupted tight junction, adherens junction and desmosome structures. We then assayed the expression of important components of junctional complexes by Western Blotting and immunostaining, including ZO-1, ZO-2, β -CATENIN, α -E-CATENIN, PLAKOGLOBIN, DESMOPLAKIN, CLAUDIN 3 and 5, and OCCLUDIN (Fig 4D-F and EV3C). Among these proteins, the tight junction protein ZO-1 lost its localization to the cell membrane in KO mice (Fig 4F). PLAKOGLOBIN, a component of desmosome and adherens junction, showed a decrease in protein expression (Fig 4D). We also measured the levels of CLAUDIN3, CLAUDIN5, and OCCLUDIN by Western blotting as the reviewer suggested. However, we did not observe any expression changes of these proteins in KO mice (Fig. EV3C). These data suggest that loss of O-GlcNAcylation may affect multiple components of junctional complexes.

Data on Fig.5D, Fig.6H,I Fig.8B are over-interpreted and must not be analysed by t-test, due to the kinetic nature of the analyses, in which points are connected to each other. A 2-ANOVA with a Bonferroni (or Holm-Sidak, more power) post-hoc test must be applied instead. According to the new results the underlined sentence "Using fecal microbiota transplantation and Paneth cell-specific deletion of OGT in mice, we demonstrate that microbial dysbiosis and Paneth cell dysfunction are insufficient to induce spontaneous inflammation, but exacerbate chemical-induced colitis" may need to be revised.

Response: We appreciate the reviewer for the critical suggestion. We performed 2-way ANOVA with Bonferroni post-hoc tests as suggested by the reviewer. However, the differences in Fig 6J and K did not reach any statistical significance. Therefore, we have changed the statement in the revised manuscript as follows: "When induced to colitis by DSS, *Defa6-Ogt* KO mice receiving the dysbiotic microbiota derived from *Vil-Ogt* KO mice had a tendency to lose more body weight (Fig 6J), show more severe colitis (Fig 6K), and express higher levels of inflammatory genes including *Ifng* and *Il1b* (Fig 6L), when compared to control mice receiving the same FMT." We also revised the Abstract accordingly.

Page 12 of the merged pdf: explain the rationale to use only 2-hour fasting for FITCdextran; please specify whether water was removed or not; the molecular weight of the dextran used is missed.

Response: Variable times in food and water fasting have been used for FITC-dextran assays [1-3]. Fasting is normally applied to avoid the interruption of gastrointestinal motility by food. We tested different pre-fasting times (0 hour, 2 hours, and overnight fasting before gavage) while no food was provided after gavage. We were able to consistently show the increased permeability in *Vil-Ogt* KO mice (Fig 4A and data not shown). Our veterinarians pointed out that long-time fasting might introduce extra stress to DSS-treated animals and suggested to use 2-hour fasting as 2-hour is the average time for the whole GI transit in mice (based on [4] and our experience). We believe these FITC-assays largely avoided the interruption of food and represented gut permeability. The average the molecular weight of the dextran is 3,000-5,000. This information has been added to the revised Method.

Page 12 of the merged pdf: in the "murine colitis model" chapter the right symbol for molecular weight is kDa (and not kD).

Response: We apologize for the typo. This has been corrected in the revised manuscript.

Referee #2:

Several previous publications have demonstrated the association between OGlcNAcylation and inflammation in several models and cell types. A reference to these previous publications could be included into the introduction, such as: Molecules. 2017 Dec 26;23(1). Clin Exp Immunol. 2017 Dec 16. Inflamm Res. 2015 Dec;64(12):943-52.

Response: Previous publications suggest that O-GlcNAcylation is associated with both pro- and anti-inflammation effects in different conditions. These references have been included in the revised manuscript and discussed in the Discussion section.

This study demonstrates that paneth cell dysfunction might be behind the increased intestinal permeability due to inhibition of O-GlcNAcylation in IECs. I think a more profound study of paneth cells would improve the quality of the manuscript: number of paneth cells in organoids, for instance, activation of autophagy at the crypt bottom.

Response: Thank you for the point and we performed more analyses on Paneth cells as the reviewer suggested. To assess Paneth cell function in vitro, we cultured ileal organoids and found that most organoids from *Defa6-Ogt* KO mice were not viable (Fig 6H), suggesting that OGlcNAcylation is indispensable for Paneth cells to provide niche factors for the growth and differentiation of intestinal stem cells in culture [5]. These niche factors such as EGF, TGF, and Wnt, however, have redundant, non-Paneth cell sources in vivo [6]. Thus, we did not observe any defects in the structure and renewal of the intestinal epithelium in *Defa6-Ogt* KO mice, despite of a significant reduction in Paneth cell numbers (Fig 6E, F).

Mounting evidence suggests that IBD-associated mutations in autophagy-related genes impair Paneth cell function [7-10]. Consistent with our recent finding that protein O-GlcNAcylation promotes hepatic autophagy [11], pro-LC3 was accumulated and failed to be cleaved and activated to form LC3-II in the IEC of *Vil-Ogt* KO mice (Fig 6D). These data indicate that loss of OGT impairs autophagy in Paneth cells.

On the other hand, two publications showed the opposite effect upon modulation of OGlcNAcylation in myeloid cells or complete knock-out mice.

1. Myeloid-derived cullin 3 promotes STAT3 phosphorylation by inhibiting OGT expression and protects against intestinal inflammation.

2. Elevated O-GlcNAcylation promotes colonic inflammation and tumorigenesis by modulating NF- κ B signaling. I agree with the authors that the cell-specific strategy is adequate in order to target a single cell type, as discussed in the manuscript. However, they did not consider the other publication into the discussion, which I think should be mentioned. On the other hand, these two publications demonstrated that STAT3 or NF κ B were the target pathways, respectively. Based on this, I would recommend the authors to provide data in order to discard these two pathways as key players in intestinal epithelium, which would improve the quality of the manuscript.

Response: We appreciate the reviewer's suggestion and agree that the effect of O-GlcNAc varies in different contexts. These two publications by Li et al. and Yang et al. demonstrated that NF- κ B and STAT3 are targets of O-GlcNAcylation. However, we did not observe any changes in STAT3 phosphorylation and NF- κ B activation in the intestine of *Vil-Ogt* KO mice (Fig 7A).

RNAsequencing and the following analyses did not identify STAT3 or NF- κ B as major potential upstream regulators of the differentially expressed genes either. We propose that protein hyper-O-GlcNAcylation in the immune system may promote immune activation and intestinal inflammation.

In intestinal epithelial cells, NF- κ B and STAT signaling pathways are protective for intestinal epithelial function, as IEC-specific deficiencies in either the NF- κ B complex or STAT family proteins cause or predispose to intestinal inflammation [12-16]. Our data suggest STAT1 as a potential downstream target of protein O-GlcNAcylation and mediator of intestinal damage in *Vil-Ogt* KO mice. Nevertheless, further studies are warranted to delineate the precise control and function of protein O-GlcNAcylation in intestinal homeostasis and inflammation. These points have

been included in the revised Discussion section.

Despite this nice link between inflammation and O-GlcNAcylation, a correlation study between clinical and expression data is needed. For instances, it would be nice to show a correlation between the degree of inflammation and the expression of OGT and OGN. Therefore, it would be nice to include more patients into the study.

Response: We appreciate the reviewer for the insightful suggestion. We obtained intestinal samples from another cohort of UC and CD patients. Immunostaining confirmed the reduction of both OGT and protein O-GlcNAcylation in UC and CD patients, compared to controls (Fig 1C). When correlating to the degree of inflammation, we were able to find that levels of both OGT and protein O-GlcNAcylation were negatively associated with inflammation statuses in UC and CD samples (Fig 1D and E). Statistical significance was reached even with limited numbers of samples, emphasizing the clinical importance of epithelial O-GlcNAcylation in intestinal pathology.

Minor comments:

Fig. 1: include severity of inflammation from human IBD patients included in the analysis

Response: The severity of inflammation was scored based on intestinal histology and was shown in Dataset EV1.

Fig. 4C: symbols indicating TJs, desmosomes and AJs are not visible.

Response: We have enlarged the symbols and changed color to white to make them easily visible. Thank you for pointing out this matter.

Fig. 6: paneth cell dysfunction; survival of paneth cells in organoid culture system, for instance

Response: As discussed earlier and shown in Fig 3H and 6H, loss of OGT in all intestinal epithelial cells or specifically Paneth cells resulted in organoid death. It demonstrated the functional importance of protein O-GlcNAcylation, but at the same made it impossible for use to directly study Paneth cell dysfunction in the organoid culture system. Nevertheless, our in vivo characterizations clearly showed increased Paneth cell death and dysfunction in *Vil-Ogt* KO and *Defa6-Ogt* KO mice (Fig 3 and 6).

Fig. 7: include STAT3 phosphorylation and NFκB activation.

Response: As shown in the new Fig 7A, we determined STAT3 phosphorylation and NF-κB activation in the colon; however, no difference between WT and KO were observed. These data suggest that OGT regulates intestinal homeostasis independently of STAT3 and NF-κB.

References:

- Alenghat, T., et al., *Histone deacetylase 3 coordinates commensal-bacteria-dependent intestinal homeostasis*. Nature, 2013. **504**(7478): p. 153-7.
- An, G., et al., *Increased susceptibility to colitis and colorectal tumors in mice lacking core 3-derived O-glycans*. J Exp Med, 2007. **204**(6): p. 1417-29.
- Hartmann, P., et al., *Deficiency of intestinal mucin-2 ameliorates experimental alcoholic liver disease in mice*. Hepatology, 2013. **58**(1): p. 108-19.
- Schwarz, R., et al., *Gastrointestinal transit times in mice and humans measured with 27Al and 19F nuclear magnetic resonance*. Magn Reson Med, 2002. **48**(2): p. 255-61.
- Barker, N., *Adult intestinal stem cells: critical drivers of epithelial homeostasis and regeneration*. Nat Rev Mol Cell Biol, 2014. **15**(1): p. 19-33.
- Shoshkes-Carmel, M., et al., *Subepithelial telocytes are an important source of Wnts that supports intestinal crypts*. Nature, 2018.
- Patel, K.K. and T.S. Stappenbeck, *Autophagy and intestinal homeostasis*. Annu Rev Physiol, 2013. **75**: p. 241-62.
- Chu, H., et al., *Gene-microbiota interactions contribute to the pathogenesis of inflammatory bowel disease*. Science, 2016. **352**(6289): p. 1116-20.
- Cadwell, K., T.S. Stappenbeck, and H.W. Virgin, *Role of autophagy and autophagy genes in*

- inflammatory bowel disease*. *Curr Top Microbiol Immunol*, 2009. **335**: p. 141-67.
10. Henckaerts, L., et al., *Genetic variation in the autophagy gene ULK1 and risk of Crohn's disease*. *Inflamm Bowel Dis*, 2011. **17**(6): p. 1392-7.
11. Ruan, H.B., et al., *Calcium-dependent O-GlcNAc signaling drives liver autophagy in adaptation to starvation*. *Genes Dev*, 2017. **31**(16): p. 1655-1665.
12. Nenci, A., et al., *Epithelial NEMO links innate immunity to chronic intestinal inflammation*. *Nature*, 2007. **446**(7135): p. 557-61.
13. Kajino-Sakamoto, R., et al., *Enterocyte-derived TAK1 signaling prevents epithelium apoptosis and the development of ileitis and colitis*. *J Immunol*, 2008. **181**(2): p. 1143-52.
14. Chiriac, M.T., et al., *Activation of Epithelial Signal Transducer and Activator of Transcription 1 by Interleukin 28 Controls Mucosal Healing in Mice With Colitis and Is Increased in Mucosa of Patients With Inflammatory Bowel Disease*. *Gastroenterology*, 2017. **153**(1): p. 123-138 e8.
15. Willson, T.A., et al., *Deletion of intestinal epithelial cell STAT3 promotes T-lymphocyte STAT3 activation and chronic colitis following acute dextran sodium sulfate injury in mice*. *Inflamm Bowel Dis*, 2013. **19**(3): p. 512-25.
16. Gilbert, S., et al., *Enterocyte STAT5 promotes mucosal wound healing via suppression of myosin light chain kinase-mediated loss of barrier function and inflammation*. *EMBO Mol Med*, 2012. **4**(2): p. 109-24.

2nd Editorial Decision

23 May 2018

Thank you for the submission of your revised manuscript to EMBO Molecular Medicine. We have now received the enclosed reports from the referees that were asked to re-assess it. As you will see the reviewers are now globally supportive and I am pleased to inform you that we will be able to accept your manuscript pending final editorial amendments.

Please submit your revised manuscript within two weeks. I look forward to seeing a revised form of your manuscript as soon as possible.

***** Reviewer's comments *****

Referee #1 (Comments on Novelty/Model System for Author):

The manuscript was high improved and both the cellular as well as the animal model strategy are very sound

Referee #1 (Remarks for Author):

all my points were well addressed.
now the manuscript has been highly improved.

Referee #2 (Remarks for Author):

The authors have successfully addressed all the critical points from the reviewers. The current version has significantly improved and contains new important information about cell specificity, signaling pathways and clinical data from the included cohort. All in all, I think the manuscript is adequate for publication.

2nd Revision - authors' response

01 June 2018

It is our great pleasure to know our revised manuscript has addressed all reviewers' concerns and is accepted in principle to be published in EMBO Molecular Medicine. We have amended the manuscript as you suggested [...].

Corresponding Author Name: Hai-Bin Ruan
Journal Submitted to: EMBO Molecular Medicine
Manuscript Number: EMM-2017-08736